



# The 2018 fire season in North America as seen by TROPOMI: aerosol layer height validation and evaluation of model-derived plume heights

Debora Griffin[1], Christopher Sioris[1], Jack Chen[1], Nolan Dickson[1,2], Andrew Kovachik[1,2], Martin de Graaf[3], Swadhin Nanda[4], Pepijn Veefkind[3,4], Enrico Dammers[1], Chris A. McLinden[1], Paul Makar[1], and Ayodeji Akingunola[1]

[1]Air Quality Research Division, Environment and Climate Change Canada, Toronto, Ontario, Canada
[2]University of Waterloo, Waterloo, Ontario, Canada
[3]Royal Netherlands Meteorological Institute (KNMI), De Bilt, The Netherlands
[4]Delft University of Technology, Delft, The Netherlands

**Correspondence:** D. Griffin (debora.griffin@canada.ca)

**Abstract.** Before the launch of TROPOMI, only two other satellite instruments were able to observe aerosol plume heights globally, MISR and CALIOP. The TROPOMI aerosol layer height is a potential game changer, since it has daily global coverage and the aerosol layer height retrieval is available in near-real time. The aerosol layer height can be useful for aviation and air quality alerts, as well as for improving air quality forecasting related to wildfires. Here, TROPOMI's aerosol layer height product is evaluated with MISR and CALIOP observations for wildfire plumes in North America for the 2018 fire season (June to August). Further, observing system simulation experiments were performed to interpret the fundamental differences between the different products. The results show that MISR and TROPOMI are, in theory, very close for aerosol profiles with single plumes. For more complex profiles with multiple plumes, however, different plume heights are retrieved: the MISR plume height represents the top layer, and the plume height retrieved with TROPOMI tends to be an average altitude of several plume layers.

The comparison between TROPOMI and MISR plume heights shows, that on average, the TROPOMI aerosol layer heights are lower, by approximately 600 m, compared to MISR which is likely due to the different measurement techniques. From the comparison to CALIOP, our results show that the TROPOMI aerosol layer height is more accurate for thicker plumes and plumes below approximately 4.5 km.

MISR and TROPOMI are further used to evaluate the plume height of Environment and Climate Change Canada's operational forecasting system FireWork with fire plume injection height estimates from the Canadian Forest Fire Emissions Prediction System (CFFEPS). The modelled plume heights are similar compared to the satellite observations, but tend to be slightly higher with average differences of 270-580 m and 60-320 m compared to TROPOMI and MISR, respectively.





# 1 Introduction

Wildfires are a significant source of air pollution, that can adversely impact the air quality in populated areas (e.g. Landis et al., 2018; Meng et al., 2019). In recent years fire behaviour has also become more aggressive in North America due to increased temperatures, drought, high fuel loading and tree death (e.g. Kitzberger et al., 2007; Littell et al., 2009; Westerling, 2016). As such, the number and size of wildfires has been shown to increase with larger areas being burned (e.g. Landis et al., 2018). Wildfires emit fine particulate matter ($PM_{2.5}$), and trace gases, including nitrogen oxides ($NO_x$), carbon monoxide (CO) and ammonia ($NH_3$) (Akagi et al., 2011; Andreae, 2019, and references therein). These traces and aerosols negatively impact air quality and are all harmful to people and their environment (e.g. Anenberg et al., 2018; Schraufnagel et al., 2019). The amounts released during the fire are highly variable and depend on the fuel type and intensity of the fire. Due to the nature of wildfires with plume heights reaching several kilometres, aerosol plumes produced by wildfires can be transported over vast distances (e.g. Damoah et al., 2004; Derwent et al., 2004; Duck et al., 2007; Lutsch et al., 2016, 2019). Plumes from larger fires can thus cause aviation hazards and affect regional air quality thousands of kilometres away from the source and even across continents (e.g. Colarco et al., 2004; Jaffe et al., 2004; Teakles et al., 2017).

With the increased fire intensity and number of fires, there is an increased necessity on modelling and forecasting smoke impacts from wildfires to be able to accurately predict the concentration of harmful pollutants, and to issue necessary alerts on time (e.g. Yue et al., 2015). The height of the smoke plumes has a large influence of where these pollutants are being transported. Thus, an extremely important component in predicting the air quality due to wildfires is to have a good understanding of the plume height and plume rise from wildfires. If the plume height and plume rise is not adequately simulated, the transport of pollutants, thus the final surface-level $PM_{2.5}$ concentration will be incorrectly modelled.

Experimentally, plume height can be measured from the ground, aircraft and space. Satellite measurements of plume heights have the advantage of superior coverage than ground-based and aircraft-borne measurements would allow. As such, satellite-remote sensing measurements are an essential tool in observing the plume heights from wildfires. So far only two satellite instruments, namely the Multi-angle Imaging SpectroRadiometer (MISR; Diner et al., 1999) and the Cloud-Aerosol Lidar with Orthogonal Polarization (CALIOP; Winker et al., 2003), were able to observe the altitude of smoke plumes on a global scale. The method used to determine the height of the plume is very different for these two instruments, thus, it is important to understand and quantify the differences of the of the plume height retrievals. MISR observes every scene from nine different angles which are then used to estimate the height of the plume. CALIOP is an active lidar instrument that can provide very detailed vertical profiles of clouds and aerosols, and can observe optically thin plumes. However, these two instruments have the disadvantage of very limited coverage where most fires are missed (Diner et al., 1999; Winker et al., 2007, 2003). The recently launched Tropospheric Monitoring Instrument (TROPOMI) can potentially fill this gap due to its daily global coverage combined with its relatively high spatial resolution. TROPOMI is a passive sensor that provides daily global coverage. TROPOMI has a new product that is dedicated to retrieval of the height of tropospheric aerosols. The TROPOMI aerosol layer





height product utilizes a very different method to than those used for MISR or CALIOP: the TROPOMI algorithm estimates the plume height based on the absorption by oxygen ($O_2$) in the $A$ band between 759 and 770 nm. A similar approach has been applied to the measurements from the Earth Polychromatic Imaging Camera (EPIC) on DSCOVR (Deep Space Climate Observatory) (Xu et al., 2017, 2019), however, this product is currently not operational and only a number of case studies are
available.

Some studies have compared MISR and CALIOP plume heights, but very few coincident overpasses exist over fires, and the time difference of approximately 2 h can create additional challenges for comparing the plume heights, as the fire is expected to increase in intensity throughout the morning with the peak fire activity being in the early afternoon (Kahn et al., 2008; Tosca
et al., 2011; Gonzalez-Alonso et al., 2019).

In this study, the TROPOMI aerosol layer height is evaluated for the first time. The aerosol layer height from three satellite instruments (MISR, CALIOP, and TROPOMI) that can measure the plume height are compared for the 2018 fire season (June-August) in North America. Finally, we also compare the satellite observed plume heights to those from Environment
and Climate Change Canada's (ECCC) air quality forecast modelling system, namely, FireWork with smoke plume injection heights based on the Canadian Forest Fire Emissions Prediction System (CFFEPS).

This paper is organized as follows: In Sect. 2, the three different satellite-borne instruments as well as the air quality model is described. Section 3 describes the effect of the different measurement techniques of the satellites on the plume height estimate
using solely modelled aerosol profiles. The inter-comparison of the three satellites plume height observations and the modelled plume height are discussed in Sects. 4 and 5, respectively. A summary and conclusions are provided in Sect. 6.

## 2 Datasets

### 2.1 TROPOMI

TROPOMI is the single payload on the Copernicus Sentinel-5P satellite that was launched on October 13, 2017. The satellite
has near full-surface coverage on a daily basis with a local overpass time of around 13:30 (Veefkind et al., 2012). The instrument has four spectrometers: three that cover the Ultra-Violet-Near Infra-Red (UV-NIR) with two spectral bands at 270-500 nm and 675-775 nm, and one for the Short-Wave Infra-Red (SWIR). The spatial resolution of TROPOMI varies with across-track position and is between $3.6 \times 7.2 \, \mathrm{km}^2$ (in UV-NIR) and $7.2 \times 7.1 \, \mathrm{km}^2$ (SWIR). As of 6 August 2019, the along-track sampling has been improved to 5.6 km.

The TROPOMI Aerosol Layer Height (AER_LH) algorithm was developed by the Royal Netherlands Meteorological Institute (KNMI) and utilizes the absorption in the oxygen $A$ band of the spectrum between 759 and 770 nm (Sanders and de Haan, 2016). The oxygen $A$ band is a highly structured line absorption spectrum with strongest absorption lines occurring between





760 and 761 nm. An aerosol layer aloft will decrease the photon path of the backscattered solar radiation, due to scattering by the aerosol layer, compared to backscattered radiation in a similar scene without the aerosol layer. This decreases the depths of the absorption lines in the oxygen $A$ band of the measurements of the scene with the aerosol layer. The aerosol layer mid height is estimated from a fit of the measurements to a simplified, single aerosol layer model simulation of the oxygen $A$ band re-

flectance, using an optimal estimation scheme, under cloud-free conditions. The final height reported is the difference between top pressure and bottom pressure of the assumed uniform scattering layer with a constant thickness of $50\,\mathrm{hPa}$. A more detailed description of the TROPOMI aerosol layer height product can be found in Nanda et al. (2019) and Sanders and de Haan (2016). Due to the importance of the backscatter signal in the retrieval, which can be dominated by the surface reflectance in case of bright surfaces and thin aerosol layers, the aerosol layer height is expected to be more robust over dark surfaces such as sea

and oceans (Sanders and de Haan, 2016).

The aerosol layer height can give insight into the height of aerosols in the free troposphere of plumes from wildfires, volcanoes, and desert dust. This product could supply important information in a timely manner on aerosol location and transport of wildfire plumes for the purpose of air quality forecasting and aviation safety.

There are two versions available for the TROPOMI aerosol layer height: the near-real-time (NRTI) product is available approximately 3 h after the satellite overpass, and the offline (OFFL) product which is available approximately 2 weeks after the satellite overpass. The algorithm for the NRTI and OFFL product is the same, however, not all products needed for the retrieval are available in NRTI. Therefore, differences between the NRTI and OFFL products include:

– In NRTI the VIIRS cloud mask is not available, and a cloud mask is constructed from the FRESCO cloud product, brightness thresholds and scene homogeneity.

– In the OFFL product a different solar irradiance spectrum may be used (if a future irradiance spectrum is closer to the radiance measurement). This should not change the results much.

In general, the OFFL product should perform better and is a better choice if timeliness is not an issue. Here, we evaluate the

OFFL version only, as the NRTI version was not available for the time period that we investigated. A quality flag is provided in the files, which is 1 if a plume is present and 0 if a plume could not be detected. Apart from the quantitative layer height, the quality flag provided alongside can be useful by itself. Just this quality flag can be useful to locate and identify presence of aerosol plumes and its vertical shape.

## 2.2   MISR

The Multi-angle Imaging SpectroRadiometer (MISR) instrument is on NASA's Terra spacecraft that has been in orbit since 1999. MISR has nine fixed push-broom cameras and views every scene from nine different angles. Each of these cameras has four line-array charge-coupled devices (CCDs) covering spectral bands centred at 446, 558, 672, and 867 nm. Its highest spatial sampling is 275 m at all angles. This design allows it to measure the height of smoke plumes using stereoscopic techniques





(Muller et al., 2002; Zakšek et al., 2013; Fisher et al., 2014; Val Martin et al., 2018). The height retrieval from MISR is not impacted by bright surfaces (Martonchik et al., 2004). MISR has approximately three overpasses daily over North America at around 10:30 am local time, with a swath width of approximately 360 km.

The plume height is not a standard product of MISR, and we used the visualization and analysis program called MISR INteractive eXplorer (MINX) tool to retrieve the plume heights (Nelson et al., 2008, 2013). This tool takes advantage of wind-direction information inherent in smoke plumes from active fires to determine plume heights and wind speeds at higher resolution and with greater accuracy than provided by the standard, operational MISR product (Kahn et al., 2007). MINX is an interactive visualization and analysis program written in IDL and designed to make MISR data more accessible to science users.

Its principal use is to retrieve heights and motion for aerosol plumes and clouds using stereoscopic methods. Within MINX, each plume has to be processed individually and plume shapes have to be digitized manually. Moderate Resolution Imaging Spectroradiometer (MODIS) brightness temperature anomalies are used to help locate the fire plumes, and plume heights are retrieved for smoke plumes close to the fire hotspots. The red-band data have a higher horizontal resolution (275 m), however, where contrast is poor within plume features and between the plume and the surface, blue-band retrievals provide better results

at 1.1 km resolution (Val Martin et al., 2018). In this study, we used the blue-band results with "good" and "fair" quality flags. Further details can be found in Kahn et al. (2007); Val Martin et al. (2010); Nelson et al. (2013). Limitations of the MISR instrument include: (1) the swath limits the global coverage, thus many smoke plumes can be missed, and (2) the local overpass time around 10:30 precedes the daytime peak in fire activity.

## 2.3   CALIOP

CALIOP, part of the Cloud-Aerosol Lidar and Infrared Pathfinder Satellite Observation (CALIPSO) satellite that was launched in 2006, is a two-wavelength (532 nm and 1064 nm) polarization-sensitive lidar. CALIOP can provide high-resolution vertical profiles of aerosols and clouds, as well as their optical properties (Winker et al., 2003, 2007). It is an active satellite instrument sensing pulses of light at 532 and 1064 nm. The back-scattered return is measured through a 1 m telescope measuring the intensities at 1064 nm and two at 532 nm (parallel and perpendicular to the polarization plane of the transmitted beam). The

vertical resolution of the cloud and aerosol profiles is between 120-360 m and the footprint is 90 m. CALIOP can detect even very thin aerosol layers with an aerosol optical thickness of 0.01 with sufficient averaging (McGill et al., 2007). CALIOP has approximately 3 overpasses at 1:30 and 13:30 local time over North America, and has a very narrow swath width of just a few kilometres. In this study, we use the daytime aerosol layer product v4 ("Layer_Top_Altitude", "Layer_Base_Altitude") (McGill et al., 2007; Vaughan et al., 2009) which provides the top and base height of aerosol layers detected (between the

surface and 30 km) averaged to a 5 km horizontal resolution, and filter out all aerosol plumes except those containing smoke or polluted dust. While CALIOP has excellent vertical resolution and has the ability to resolve the layer heights of multiple plumes in a single profile, its swath width is very narrow and has a 16-day global coverage.



## 2.4 MODIS

The MODIS thermal anomaly product (MOD14) (Giglio et al., 2003, 2006, 2016) is used here to locate the wildfires. There are currently two MODIS instruments in space, on NASA's Terra and on NASA's Aqua satellites. Daytime measurements of TERRA and AQUA are around 10:30 and 13:30 local time, respectively. Here, we utilized the thermal anomalies for both

MODIS instruments. Note, that fires can potentially be missed due to cloud cover.

## 2.5 GEM-MACH

We also make use of the satellite-derived plume heights to evaluate the modelled plume heights from an experimental version of ECCC's FireWork biomass burning air quality forecast modelling system. The core of the FireWork system is the Global Environmental Multiscale - Modelling Air-quality and Chemistry (GEM-MACH) coupled meteorology and chemical trans-

port model. GEM-MACH contains a detailed representation of atmospheric chemistry, including emissions, dispersion, and removal processes of 42 gaseous and 8 particle species, which reside within the physics module of the Global Environmental Multiscale (GEM) weather forecast model (Côté et al., 1998; Girard et al., 2014). The operational version of the model (Moran et al., 2010; Pendlebury et al., 2018) has a horizontal resolution of $10 \times 10\,\mathrm{km}^2$ for the North American domain and 80 vertical levels (from the surface to approximately 0.1 hPa) on a hybrid pressure grid. The forecast system produce air quality forecast

conditions for 48-hours and is initialized every 12 hours at 00 and 12 UTC.

The experimental GEM-MACH system was used as part of an ensemble of models contributing to the FIREX-AQ experiment at a resolution of 2.5km – here, the same system was used at 10km resolution, to simulate forest fire emissions, transformation and transport for the summer of 2018 (1 June to 31 August 2018), with an internal model "physics" time step

of 7.5 minutes, and output provided every hour. The outputs for the simulations included $PM_{2.5}$ fields, and estimates of the aerosol optical depth at a variety of wavelengths calculated internally using an on-line Mie lookup table (Makar et al., 2015b, a).

Near-real time fire hotspot information is obtained from three satellite sensors: MODIS, the Advanced Very High Resolution Radiometer (AVHRR), and Visible Infrared Imaging Radiometer Suite (VIIRS) processed through the Canadian Wildland

Fire Information System operated by the Canadian Forest Service, Natural Resources Canada (http://cwfis.cfs.nrcan.gc.ca, last access: 1 October 2019 Lee et al., 2002). Hourly fire emissions and smoke plume injection heights were estimated with the CFFEPS module at individual hotspot location. Further details describing the implementation of the GEM-MACH wildfire component within the model can be found in e.g. Munoz-Alpizar et al. (2017); Pavlovic et al. (2016); Chen et al. (2019).

Previous work with CFFEPSv2.03 (Chen et al., 2019) showed a substantial improvement in forecast skill for daily maximum values of particulate matter, $NO_2$ and $PM_{2.5}$ relative to the previous ECCC operational forecast which employed a much simpler Briggs plume rise approach for forest fire emissions plume rise. Here, we investigate how the particulate mass and plume injection height calculated with GEM-MACH and from CFFEPSv4.0 and transported downwind over a short period





of time by GEM-MACH near fire hotspot locations compares to satellite-derived plume heights. In order to allow a direct comparison between satellite-derived plume heights and those predicted by GEM-MACH/CFFEPSv4.0, the hourly modelled $PM_{2.5}$ concentrations were interpolated temporally to the satellite overpass times. Only plumes due to fires are investigated: we subtracted the model run without fire emissions from the run with fire emissions to remove $PM_{2.5}$ contributions from none-fire
sources.

## 3    Observing System Simulation Experiments (OSSE)

The three satellite instruments are fundamentally very different and use three different parts of the radiative spectrum to determine the plume height. Here, we evaluate simulated plume heights from model output using similar techniques as MISR and TROPOMI, respectively, for several modelled aerosol profiles. This will help to interpret the fundamental differences
between these retrieval techniques and to confirm whether the satellites are observing the same plume and to evaluate the methodology for model plume height estimation best suited for comparison to the satellite-derived plume heights. The aerosol profiles used here are 24 modelled profiles (from the GEM-MACH model) containing smoke at various altitudes between approximately 1 and 7 km with various Aerosol Optical Depths (AODs). Nine example profiles of these 24 are shown in Fig. 1d-l. Note, this section is using only modelled aerosol profiles (no satellite observations were used here) with the aim of
evaluating the different retrieval algorithms and understanding what "simple" plume height definitions can be used to compare the model output to the satellite observations.

### 3.1    OSSE MISR plume heights

In order to simulate the layer height retrieved by MISR using aerosol profiles from GEM-MACH, we rely on the concept that MISR's layer height is defined as the layer contributing the most to the reflective contrast relative to the surrounding air (Kahn
et al., 2007). Thus, to determine the MISR-equivalent plume layer height from the GEM-MACH profiles, we simply calculate the dI/dNz weighting function where I is the 672 nm monochromatic radiance at the top-of-the-atmosphere for a viewing zenith angle of 26°. Nz is the GEM-MACH aerosol number density at altitude z and the weighting function is calculated numerically by perturbing layers of the profile independently and determining the radiance difference relative to the unperturbed case. The $PM_{2.5}$ aerosol number density vertical profile is obtained from GEM-MACH for these smoke cases. The number density is
obtained from the model's fine-mode mass density profile, assuming a typical mass of a fine-mode particle of $1.55 \times 10^{-9}$ µg based on a particle density of 1.35 g/cm$^3$ (Reid and Hobbs, 1998), and assuming spherical particles with a radius of 130 nm. These approximations used here may not necessarily reflect GEM-MACH's predictions for particulate mass, radius or density, but those assumptions have been used to reflect that smoke particles tend to be small and to make the interpretation of the results less complicated by using the same assumptions for each simulated case. The retrieved layer heights will not depend
on a multitude of aerosol properties. The VECTOR radiative transfer (RT) model is used (McLinden et al., 2002) and aerosol scattering is simulated using Mie theory. For the Mie calculations, a gamma distribution is used for the aerosol size distribution (Eq. 2.56 of Hansen and Travis (1974)) with an effective radius of 130 nm and an effective variance of 130 nm and a size range



of 0.01 to 260 nm. The complex refractive index is appropriate for external mixed black carbon at 99 % relative humidity: 1.68+0.36i (Kou, 1996). Note, that this might not reflect the true aerosol size distribution of a fire smoke plume. However, the approximation can be used since the retrieved layer contributing the most to the reflective contrast does not depend on the exact size distribution used. The surface albedo provided in the TROPOMI layer height product is used for each different scene

(for MISR and TROPOMI) and five orders of scattering were computed. For the MISR and the TROPOMI OSSE (see below), it is critical to have fine layering in the radiative transfer model simulations that serve as the pseudo-observations in order to properly capture the shape of GEM-MACH aerosol profile. For MISR simulations using VECTOR, 100 m layering was used in the lowest 20 km of the atmosphere and, thus, the GEM-MACH aerosol profiles were interpolated to 100 m layers.

## 3.2    OSSE TROPOMI plume heights

To simulate the layer height retrieval from TROPOMI, we used MODTRAN 5.2 (Berk, 2013, and references therein) to take advantage of the correlated-k option for simulating radiances in an absorption band, namely the oxygen $A$ band ($\sim$762 nm). The correlated-k absorption parameter data are specified at 1 cm$^{-1}$ resolution. The terrain height for the MODTRAN modelling is obtained from GEM-MACH for each scene. The radiance is convolved with a triangular slit function with a full width at half maximum of 9 cm$^{-1}$ to account for the TROPOMI spectral resolution in channel 6 (Veefkind et al., 2012), which covers

the O$_2$ $A$ band. The discrete ordinates method is used to simulate the radiative transfer with 8 streams. The solar zenith angle and viewing nadir angle of each scene is taken into account (as was done for MISR-OSSE). MODTRAN expects an aerosol extinction profile as an input rather than an aerosol number density profile. This conversion involves scaling the number density profile determined in Sect. 3.1 such that the aerosol optical depth simulated by MODTRAN was equal to the aerosol optical depth simulated for MISR. The $A$ band absorption depth is used as the observable in the retrieval and is computed

using the following ratio: $(I_{13107} + I_{13145})/(I_{13005} + I_{13007} + I_{13175})$, where the sub-scripted numbers are the wavenumbers at which spectral radiances are simulated. The numerator is the sum of the radiance at two wavelengths for which O$_2$ is strongly absorbing and the denominator contains three wavelengths in the continuum (i.e. minimal absorption). The retrieval method is iterative and seeks to match the "observed" absorption depth with the forward modelled one by solely varying the layer height during the retrieval. The "observations" involve using the GEM-MACH aerosol profile, whereas for the forward model

simulations during the retrieval, the profile shape is not known and we assume that the aerosol layer has a vertical extent of 500 m with no aerosol outside this 500 m layer. The here reported layer height is the middle of this layer.

## 3.3    Plume height evaluation using pseudo-observations

In this section, the modelled plume heights, derived using five "simple" methods and the simulated plume height using modelled profiles with the MISR (Sect. 3.1) and TROPOMI (Sect. 3.2) retrieval methods, are compared. In Sect. 3.1 and 3.2, we

described methods based on remote sensing for plume height estimation using modelled aerosol profiles. These simulations are, however, time consuming and therefore not practical for the model-satellite comparison as thousands of aerosol profiles would have to be simulated. Instead, several simpler methods are considered to define plume heights from model output, that can be used to compare the modelled output to satellite observations in the subsequent section. These methods include: (1)





the altitude of the model layer of the maximum $PM_{2.5}$ concentration (shown as red down-pointing triangles in Fig. 1), (2) the altitude of the highest model layer that exceeds $PM_{2.5}$ concentration of $10\,\mu g/m^3$ (shown as blue down-pointing triangles in Fig. 1), (3) the altitude of the highest model layer that exceeds $10\,\%$ of the maximum $PM_{2.5}$ concentration (this definition has previously been used in Raffuse et al. (2012); shown as black dots in Fig. 1), (4) the average height between method (1) and

(2) (shown as cyan right-pointing triangles in Fig. 1), and (5) a $PM_{2.5}$ concentration weighted average of model layer heights (shown as magenta left-pointing triangles in Fig. 1). The results of this simulated plume height comparison are shown in Fig. 1 with the reference 1:1 line shown as a black-dash. These results show that the methodology in which the top layer of the plume is that exceeds $10\,\mu g/m^3$, method (2), is closest to the MISR simulated plume heights (Fig. 1a) with a mean difference ($\pm$ standard error) of $-98\,m$ ($\pm181\,m$). Method (3) overestimates the plume height consistently for all plumes. Method (1),

(2), (4), and (5) are very close for many aerosol profiles, but for profiles containing multiple plumes, method (2), (4), and (5) underestimate the "MISR"-simulated plume height. For the TROPOMI-OSSE (Fig 1b), simulated plume heights with method (4) is the closest with a mean difference of $37\pm90\,m$ and except two profiles, the differences are all less than $200\,m$. For simple plumes with one strong aerosol peak (Fig. 1 d-h), method (2) is close to the simulated TROPOMI-OSSE plume height, but tends to overestimate the plume height of more complicated plumes with multiple aerosol layers, while method (1) and (5)

tend to underestimate the TROPOMI-OSSE plume height. Using method (3), the altitude of $10\,\%$ of the maximum enhancement, overestimates the plume height for all plumes. Lastly, the simulated plume heights using the MISR and the TROPOMI approaches are compared over different AOD simulated using the VECTOR RT model (Fig. 1c). Overall, the plume heights estimated using the five different "simple" methods are consistent with the satellite retrievals for most plumes, however, there are cases where the TROPOMI-OSSE plume heights are lower in comparison to the MISR-OSSE plume heights. We have found

these to be unrelated to the AOD of the plume. The average mean difference ($\pm$ standard deviation) between the simulated aerosol layer heights between MISR and TROPOMI is $0.52\pm0.84\,km$. This difference can be attributed purely to the different observation/retrieval methods of the aerosol layer height between the two instruments.

The differences between the MISR-OSSE and TROPOMI-OSSE plume height was further investigated and Figs. 1d-l show

examples of the profiles used, along with the retrieved plume heights. Profiles for which the MISR-OSSE and TROPOMI-OSSE plume heights agree well are displayed in Figs. 1d-h, and all show one single dominant plume. Profiles that result in differences between MISR-OSSE and TROPOMI-OSSE are more complicated profiles consisting of multiple aerosol layers (Figs. 1i-l). In these cases, MISR observes the altitude of the upper plume, whereas the $A$ band method used for TROPOMI (and EPIC) retrieves an optical centroid altitude (Xu et al., 2019). Note that retrieving a single layer height can be difficult

particularly when the volume enclosing the plume takes the shape of a column or when there are multiple plumes at different altitudes either due to multiple source locations (i.e. points of origin) or due to shifts in wind direction or atmospheric stability during the course of emissions. Large differences between TROPOMI and MISR might be an indicator that multiple plumes are present.



## 4   TROPOMI aerosol layer heights

### 4.1   Comparison to MISR

In total, we found 115 fire plumes for which the MISR layer height retrieval was of good (87) to fair (29) quality and which were captured by both MISR and TROPOMI between June and August 2018 in North America. Most of the plumes were
located in western Canada and western U.S. where fire activity was high to extreme for the year. There were few plumes in eastern Canada in provinces of Ontario and Quebec, as well as in the states Wyoming and Colorado in the central U.S. Due to the differences in sensor spatial resolution, each plume spanned many pixels, on the order of a several hundred for MISR and a dozen for TROPOMI. For the comparison, we investigated the maximum plume heights and the mean plume heights within those fire plumes. An example is shown in Fig. 2 for two fires (the fire hotspot is shown as red dots) in central British
Columbia on 6 August 2018 at approximately 53°N, 126°W. The plume height pixels from MISR, TROPOMI and GEM-MACH are overlaid on the VIIRS true colour visible imagery showing the smoke plume (obtained from NASA Worldview, https://worldview.earthdata.nasa.gov/). For MISR a plume has to be digitally outlined in MINX (dashed red line in Fig. 2), this plume polygon was also used to define the spatial extend of the same smoke plume for TROPOMI. As MISR overpasses a location approximately 2 h earlier than TROPOMI, MISR and TROPOMI do not observe the fires at exactly the same time. Forest
fire emissions typically follow a diurnal cycle with a decrease in emissions and intensity during the night and increase throughout the day until the late afternoon - hence the plume might be expected to grow between the two overpass times. To account for plume growth from atmospheric dispersion over this time, the plume shape derived for the MISR analysis was increased spatially in size by 0.15° in longitude/latitude for TROPOMI (see navy dashed line in Fig. 2). All pixels within this slightly enlarged plume outline were assumed to belong to the same fire plume, and the mean and the maximum of those observations
were calculated for comparison with MISR. The enlarged polygon is used as a guidance which pixels from TROPOMI belong to the same plume that outlined in MINX; there is no manual input or outlining required for the TROPOMI algorithm. If the enlarged polygon is too large or the plume doesn't cover the whole area, the mean will not be affected as the TROPOMI plume heights are set to a fill value (and masked) if no plume has been detected or retrieval did not pass the quality control. Since the resolution of the MISR pixels is around $1\,\text{km}^2$ and much finer resolution than that of TROPOMI ($5 \times 7\,\text{km}^2$), greater variability
and extremes in plume heights are expected from MISR with significantly higher sampling of pixel within the same plume, as spatial smoothing of layer height is limited. To correct the impact of sensor resolution on the maximum plume height derived from a cluster of pixels in a given plume, the MISR pixels were averaged and binned on a $0.05° \times 0.05°$ grid to approximately match the TROPOMI resolution.

The results of the comparison between the TROPOMI and MISR derived plume heights for 155 identified co-locating fire plumes from both sensors in North America are shown in Fig. 3. The average maximum plume heights above ground level for the 2018 fires in North America are, on average, $2\,\text{km}$ (ranging between 0.4 and $5.5\,\text{km}$) and $1.6\,\text{km}$ (ranging between 0.01 and $8.4\,\text{km}$) for MISR and TROPOMI, respectively. The mean plume heights (above ground level) from the 155 fire plumes are on average $1.4\,\text{km}$ (ranging between 0.3 and $3.2\,\text{km}$ for MISR) and $0.8\,\text{km}$ (ranging between 0.01 and $2.8\,\text{km}$ for





TROPOMI). Overall, TROPOMI's maximum and mean plume height is on average $0.59 \pm 1.3$ km and $0.55 \pm 0.74$ km lower than the plume height derived from MISR, respectively, when horizontal resolution impacts have been removed by averaging, as noted above. The mean difference found for the TROPOMI and MISR observed plume heights is similar as found for the simulated plume heights of the OSSE. The maximum plume heights from all smoke plumes are similar, however, have a large

spread ($\sigma = 1.3$ km) and only a moderate correlation ($R = 0.44$), even when taking the difference in resolution into account by binning the data. This is expected and in fact the results are reasonable, since the maximum plume height will only contain the observations of a single TROPOMI pixel and there is a time difference between 0.5 and 3 h between the TROPOMI and MISR overpass in which plume heights can change significantly. The average plume heights, a more aggregated quantity, have a better correlation with a correlation coefficient, $R$, $R = 0.61$ and slope of best fit, $s$, $s = 0.8$, with TROPOMI biased low. This low

bias of the TROPOMI observations of plume heights is expected based on the retrieval technique (see Sect. 3), where MISR observes the top plume height and TROPOMI observes an average plume height when multiple layers of aerosols are present. Furthermore, despite the spatial adjustment of expanding the sampling footprint of MISR plume, the 0.5 - 3 h earlier overpass time of MISR is likely sampling plume heights earlier in the day when planetary boundary layer (PBL) is not fully established, and of lower fire intensity. TROPOMI plume height observations are, therefore, expected to be slightly higher compared to

MISR, because of generally increases with fire intensity in the afternoon enhancing plume advection. However, the differences between the satellite observations and the differences of the OSSE simulated plume heights based on the satellites retrieval algorithm (Sect. 3) are similar, no increasing plume height is apparent from this TROPOMI and MISR dataset.

The regional distribution of the different plume heights are illustrated in a map over locations of fire hotspots during the

analysis period (see Fig. 3 d). Table 1 summarizes the different plume heights found by MISR in comparison to TROPOMI. The average plume heights for the maximum and mean plume height within each of the 115 plumes are shown for fires in different types of biome as classified by the International Geosphere-Biosphere Programme (IGBP). Only enough observations within our dataset were found for evergreen fires for the comparison with MISR. To be able to do a quantitative regional comparison for additional vegetation types, more smoke plume observations are required.

## 4.2  Comparison to CALIOP

For the comparison between CALIOP and TROPOMI, only CALIOP profiles over North America that are flagged ("Feature_Classification_Flags") to contain smoke or polluted aerosols were selected. The maximum and mean of the TROPOMI aerosol layer height within $\pm 0.15°$ ($\sim 15$ km) of those profiles were compared. The height of the plume top and plume base are contained in the CALIOP L2 product (aerosol layer product v4) and those are on an averaged horizontal resolution of 5 km,

similar to that of TROPOMI and thus no additional corrections to the sampling footprint were carried out. On average there is a small time difference between these two sensors varying between -1 and 2 h (CALIOP-TROPOMI overpass) for this dataset, so the forest fire plume height comparisons may also be affected by plume evolution between overpasses. Unlike TROPOMI that provides one plume height at each sampling pixel, the active lidar on CALIOP provides detail plume profile, some with multiple layers of aerosol in a profile. Here, we define the thickness of the plume as the difference between plume top and





plume base; and CALIOP's mid layer height (average between plume top and base) are compared to TROPOMI's aerosol mid layer heights. We further found that very high plumes (>8 km) observed by CALIOP were not captured by TROPOMI, likely because they are optically quite thin, and those have been removed from the comparison. Sometimes, multiple layers of aerosols can occur in a CALIOP profile. We investigated additional CALIOP plume height interpretations to find the most

representative layer height for the comparison to TROPOMI. They are: (1) the CALIOP top layer height, (2) the average of all CALIOP-identified aerosol layers, and (3) the thickest (geometrical thickness) CALIOP aerosol layer. Overall, we found that the first of these definitions is not appropriate for the TROPOMI comparison as the top aerosol layer in CALIOP can be a very thin plume in the upper troposphere/lower stratosphere that is not captured by TROPOMI. The second methodology comparing CALIOP average of all aerosol plumes to TROPOMI was sometimes also biased by very low concentration layers of CALIOP

aerosol at high elevations. The third methodology was not affected by the issues for the other two methods; using comparing the CALIOP geometrically thickest aerosol layer with the TROPOMI aerosol layer height seems the most applicable for the plume height comparison between those two different satellite instruments.

     Figure 4 summarizes the CALIOP-TROPOMI plume height comparison for (a) thick plumes (>1.5 km), (b) thin plumes, (c)

a histogram of the differences, and (d) how the statistics of the comparison change for different plume thickness filters. The results show that the difference between the plume height observed by TROPOMI and CALIOP depends significantly on the thickness of the plume (as derived from CALIOP). Thicker plumes seem to be better captured by TROPOMI and the thicker the plume the smaller the difference between the CALIOP and TROPOMI plume height. TROPOMI was biased low, on average by 2.1 km, in comparison to CALIOP for thin smoke plumes (thickness of less than 1.5 km). Much better agreement and a

improved correlation between the two satellite datasets is found for thicker plumes (see Fig. 4d). The mean difference reduces with the thickness of the plumes, the mean difference between the TROPOMI and CALIOP mid aerosol layer is just 50 m for very thick plumes (>3 km). The geometrically thick plumes are typically optically thicker plumes, too. The reason for the reduced bias with increasing layer thickness is probably the sensitivity of the TROPOMI AER_LH algorithm to the scattering layer in the scene, which is more and more dominated by the surface if the aerosol layer is optically thinner. The correlation plot

and histogram are shown in Fig. 4 for thin plumes (shown in blue) and thick plumes (>1.5 km; shown in red). The distribution of the differences between the TROPOMI and CALIOP plume height is a normal distribution with a smaller spread for thick plumes. From this analysis it also appears that lower plumes, below approximately 4-4.5 km, are better captured by TROPOMI (see Fig. 4).

## 5  Model plume height evaluation

In order to compare the FireWork model plume heights to the satellite observations, the model hourly output is interpolated to the time of the satellite overpass. The mean and maximum plume heights within individual fire plumes were compared. As mentioned in Sect. 2.5 MISR and TROPOMI detects smoke plume height differently, thus, the model plume height extracted for comparison are also different. For the comparison with the MISR plume heights, the model plume height is defined as the





highest model layer containing a $PM_{2.5}$ concentration greater than $10\,\mu g/m^3$. For the comparison to TROPOMI, the model plume height was defined as the average height between the altitude of the maximum $PM_{2.5}$ concentration within the grid column and the highest layer containing a $PM_{2.5}$ concentration exceeding $10\,\mu g/m^3$.

## 5.1 Comparison with MISR

Similar as for the TROPOMI-MISR comparison, the area of the plume is defined by the expanded MISR plume outline (by $0.1°$) and all points within this enlarged polygon (an example can be seen in Fig 2 - the enlarged polygon used for the model comparison is shown as a blue dashed line) are considered as part of the plume. Given that FireWork is a forecast product, this expanded polygon is used for the comparison to compensate for errors in wind forecast speeds and direction within the model and for uncertainties related to temporal interpolation between hourly output and satellite overpass. Furthermore, given

the coarse model resolution compared to MISR pixel, the expanded plume footprint allows for more points to be extracted for comparison. All points with elevated $PM_{2.5}$ within this extended polygon are considered part of the same plume. Again, to account for the difference in resolution when comparing the maximum plume height, MISR pixels have been binned to $0.1x0.1°$ to the approximate resolution of the model.

We found that the modelled plume heights are very similar, but on average slightly higher than the ones observed by MISR. Overall, the modelled plume heights represent the observations very well in terms of mean and maximum plume heights with differences of $-0.06 \pm 0.68\,$km and $-0.32 \pm 1.21\,$km, respectively. Figure 5 summarizes the results for the comparison between MISR and CFFEPSv4.0 (a-c). In total 70 fire plumes were compared (all between June-August 2018) in terms of (a) maximum plume height and (b) mean plume height. A map illustrating the regional distribution of the mean plume heights is

shown in Fig. 5d.

The FireWork modelled plume heights with forecast meteorology are on average less than $100\,$m higher compared to the MISR observations. The modelled plume heights correlate well with the satellite observations with $R = 0.73$ for the mean plume heights. The maximum plume height within one plume is also well represented with a correlation coefficient

of $R = 0.53$, the model overestimates the maximum plume height on average by $+0.32\,$km. These are very encouraging results for modelled versus satellite-observed plume heights, especially, considering the assumptions that were parametrized in the modelling fire plume height, such as amount of fuel consumed, area burned, energy released, modelled atmospheric profiles and dispersion. Not only are the mean differences small but there is a good correlation between the observations and the model for both mean and maximum plume heights. The error difference for the plume plumes analysed here have a normal distribu-

tion (see Fig. 5c). Significant of progress has been made in recent years in terms of modelling plume rise for biomass burning. For example, Raffuse et al. (2012) found that on average the modelled plume heights agreed with the observations, but correlations between observed and simulated plume heights was poor. However, model plume heights in Raffuse et al. (2012) were calculated using a Briggs plume rise approach as opposed to calculating the energy balance in multiple atmospheric layers. The latter approach, used in CFFEPS, was found to result in more accurate predictions of surface daily maximum $PM_{2.5}$, $NO_2$





and O$_3$ than the use of Briggs formula (Chen et al., 2019). At least part of the improved model predictive performance of the ECCC FireWork forecast may be attributable to these radiative transfer calculations within CFFEPS, with the version used here (v4.0) also including a higher vertical resolution than the v2.03 version described in Chen et al. (2019). The differences between MISR and FireWork modelled plume heights for different biomes are summarized in Table 1 showing that for evergreen

forest fires the modelled and MISR observations are well on average. There were not enough fire plume available from the FireWork-MISR comparison to compare other biomes. The number of fire plumes that have been compared to the FireWork is slightly lower than for the TROPOMI-MISR comparison, some smaller fires can be missed by the model or the modelled aerosol concentration does not reach $10\,\mu$g/m$^3$.

## 5.2 Comparison with TROPOMI

For the comparison of FireWork modelled plume height to TROPOMI, the spatial extend of the plume is defined as the polygon surrounding TROPOMI's minimum and maximum latitude and longitude (shown as a purple dashed line in Fig. 2) in which a predefined smoke or aerosol layer was present near a fire hotspot as identified by MODIS Aqua. Similar to the process with MISR comparison, this polygon is then increased by $0.1°$ (black dashed line in Fig. 2) to account for the errors in forecast wind direction and speed within the model. As the resolution of TROPOMI is higher than the resolution of the GEM-MACH

model, for the maximum plume height comparison the TROPOMI observations are binned to $0.1° \times 0.1°$, approximately the resolution of the model and the TROPOMI gridded data is paired with model output interpolated to the TROPOMI over-pass time. Overall for North America June-August 2018, 671 coincident fire plumes were found for the model comparison. This number is significantly higher than for the comparison between FireWork and MISR, because of the better geospatial coverage of TROPOMI compared to MISR and, thus, less fires are missed by TROPOMI. The results for the TROPOMI and model

comparison are shown in Fig. 6 and the results summarizing the averages for different biomes are in Table 1.

Moderate correlation was found for the TROPOMI-model comparison ($R$ in the 0.3 and 0.5 range; see Fig. 6). The model plume height is on average higher than the TROPOMI observations. The average difference (TROPOMI-model) of the maximum and mean plume height is of $-0.27\pm1.84$ km and $-0.58\pm0.85$ km for the maximum and mean plume height, respectively.

For the plume heights for different biomes, also with the increased number of fire plumes with TROPOMI, only minor differences are observed between the different biomes. Overall, it seems that CFFEPS struggles the most with grassland fires where the average plume height is about 0.8 km higher than the TROPOMI observations. Plume heights from evergreens and woody savannas seem to agree well with the observations. Looking at the TROPOMI plume height, fire plumes from all here

presented biomes have on average a maximum plume height between 2.1 and 2.3 km, and an average mean plume height of 0.7 km.





## 6    Summary and Conclusions

We compared wildfire plume heights from TROPOMI and MISR-derived plume heights and CALIOP aerosol profiles, for the 2018 fire season in North America (June to August). The only satellites that could globally observe plume heights before the launch of TROPOMI were MISR and CALIOP. MISR and CALIOP are unique in their ability to vertically resolve the atmospheric aerosols globally, however, those two satellites have a narrow-swath with a global coverage every 3 and 16 days, respectively. This means that many fire plumes are missed by these satellites. The plume height product from TROPOMI is a potential game changer in terms of frequency and availability of observations: aerosol plume heights from TROPOMI have the advantage of daily global coverage and a NRTI version exists that is available approximately 3 h after the overpass. CALIOP aerosol profiles are available with an approximately one-day delay, but MISR-derived plume heights on the other hand require time consuming manual input and are not available NRTI. As such, TROPOMI aerosol layer heights can provide value to the modelling communities for improving air quality forecasting and providing improved air quality and aviation warnings, as less fires will be missed.

We simulated MISR and TROPOMI aerosol layer heights (OSSE) from different aerosol profiles to better understand the differences between the two algorithms.The plume heights for profiles with a single aerosol peak agreed almost perfectly and the aerosol layer heights from TROPOMI-OSSE and MISR-OSSE were within a just few meters. However, this is not the case for profiles with multiple aerosol layers. From the plume height retrieval using the oxygen $A$ band, the TROPOMI aerosol layer height tends to lie in between those aerosol layers. This is a significant limitation since the exact plume heights will remain unknown in cases where multiple aerosol layers are present in one profile. MISR on the other hand tends to respond to the upper aerosol layer, if there are any layers beneath MISR will not be able to pick this up. Based on our OSSE, the different retrieval techniques of TROPOMI and MISR will result in differences of 520±840 m (based on 24 profiles), with TROPOMI typically returning lower plume heights. We found a very similar bias when comparing the actual satellite observations: the TROPOMI aerosol layer heights seem to be on average approximately 600 m lower compared to the MISR plume heights. We further found, by comparing with the CALIOP aerosol profiles, that the TROPOMI aerosol layer heights are more accurate for thicker plumes: the difference between the CALIOP and TROPOMI mid-plume height decreases and the correlation increases with increasing thickness of the plume and for a 3 km thick plume the average difference is only about 50 m. Plumes below 4.5 km are better retrieved with TROPOMI. Furthermore, very high (>8 km) or thin plumes can be missed by TROPOMI.

The satellite observations have been compared to the GEM-MACH model with input from CFFEPS. From the OSSE, we found that the top altitude with $PM_{2.5}$ >10μg/m$^3$ agrees best with the MISR-OSSE ($-98 \pm 181$ m). On the other hand, TROPOMI-OSSE plume heights agree best with average between the altitude of the maximum and the top altitude with $PM_{2.5}$ >10μg/m$^3$ (37±90 m). The comparison between the model and the satellite observation shows that the simulated plume heights with CFFEPS tend to be 60-580 m higher than the observed plume heights by MISR and TROPOMI. The biggest differences between CFFEPS and the TROPOMI observations were found for plumes from grassland fires, CFFEPS overestimates the



plume height on average by nearly 1 km. With correlation coefficients between $R = 0.28$ to $R = 0.73$ between the satellite observations and the model, this is an encouraging result for modelled plume heights, as fire plumes are extremely variable and, as such, difficult to estimate and many assumptions were made to model the plume injection height.

Overall, TROPOMI aerosol layer height has been compared to MISR and CALIOP plume heights, showing moderate correlation and agreement. The TROPOMI plume heights seems more accurate for thicker and lower plumes plumes (<4.5 km altitude). TROPOMI aerosol layer height seems to be biased low, this was seen for both, the comparison to MISR and CALIOP, and is likely due to the TROPOMI measurement technique's tendency to return an intermediate plume height if multiple aerosol layers are present. Also, the TROPOMI algorithm is sensitive to the surface, which will bias the retrievals low, especially for
optically thin plumes (and bright surfaces) (Sanders and de Haan, 2016). Using the oxygen $A$ band to retrieve the aerosol layer has significant limitations if multiple smoke layers are present, leading to an average plume height. This might limit its application for aviation safety as the exact altitude of the plume may be inaccurate. However, it is still a very valuable product if one plume dominates the profile, as well as for model comparison and to enhance model performance. The product can also be useful for satellite emission estimates from wildfires, where the approximate layer height of the plume needs to be known
to get an accurate wind component of for the plume transport (e.g. Fioletov et al., 2015; Nassar et al., 2017; Adams et al., 2019; Dammers et al., 2019). For these estimates, the aerosol layer height can provide an approximate height of the plume. No significant dependencies of the fire classification, fire radiative power, albedo, or the TROPOMI solar and viewing zenith angles towards plume height estimates were found within this study, however, more data are needed for a more qualitative comparison.

*Author contributions.* D.G. compiled the analysis; C.S. performed the observing system simulations; J.C., P.M., and A.A. worked on the development of CFFEPS as well as providing feedback on the methodology used within the manuscript; M.dG., P.V., and S.N. worked on the development of the TROPOMI AER_LH product; N.D., A.K., E.D., helped develop and optimize the analysis codes. The publication was prepared by D.G., and all authors reviewed the manuscript and contributed to the discussion of the paper.

*Competing interests.* The authors have no competing interests.

*Acknowledgements.* This work contains modified Copernicus Sentinel data. The Sentinel 5 Precursor TROPOMI Level 2 product is developed with funding from the Netherlands Space Office (NSO) and processed with funding from the European Space Agency (ESA). TROPOMI data can be downloaded from https://s5phub.copernicus.eu. The MISR data used in this paper were obtained from the NASA Langley Research Center Atmospheric Science Data Center. Development of the MINX software is supported by the NASA Earth Observing System's MISR Project. We would like to thank the MISR project team for making the MINX software available. We acknowledge the use



of imagery from the NASA Worldview application (https://worldview.earthdata.nasa.gov), part of the NASA Earth Observing System Data and Information System (EOSDIS).



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



**Table 1.** Summary of plume heights observed in different regions, shown is the mean (standard deviation) of the maximum and mean plume heights for all plumes for different types of wildfires (Biome IGBP). The reported altitudes are all in km above ground level.

| Biome name (class) | # plumes | Maximum (km) | | Mean (km) | |
|---|---|---|---|---|---|
| | | MISR | TROP | MISR | TROP |
| All | 115 | 2.0 (0.09) | 1.4 (0.05) | 1.4 (0.06) | 0.7 (0.11) |
| Evergreen (1) | 84 | 1.8 (0.09) | 1.4 (0.05) | 1.3 (0.06) | 0.8 (0.09) |
| | | MISR | CFFEPS | MISR | CFFEPS |
| All | 70 | 1.7 (0.9) | 2.0 (1.0) | 1.3 (0.6) | 1.3 (0.4) |
| Evergreen (1) | 25 | 1.6 (1.0) | 1.6 (0.6) | 1.2 (0.6) | 1.1 (0.4) |
| | | TROP | CFFEPS | TROP | CFFEPS |
| All | 671 | 2.2 (1.6) | 2.5 (1.2) | 0.7 (0.5) | 1.1 (0.6) |
| Evergreen (1) | 263 | 2.1 (1.3) | 2.3 (0.9) | 0.7 (0.4) | 1.1 (0.6) |
| Woody savannas (8) | 197 | 2.3 (1.8) | 2.3 (1.0) | 0.7 (0.4) | 1.1 (0.6) |
| Grassland (10) | 136 | 2.2 (1.7) | 3.0 (1.8) | 0.7 (0.5) | 1.5 (0.9) |



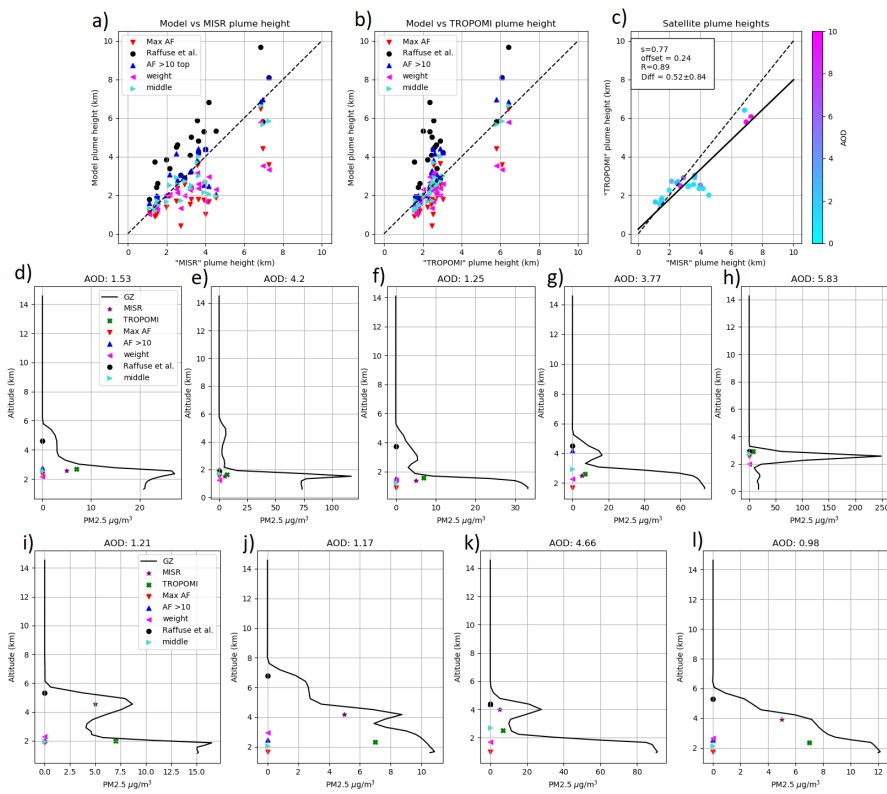

**Figure 1.** Observing System Simulation Experiments (OSSE) results for the simple methods (see text) versus simulated-MISR (a) and simulated TROPOMI heights (b). Simulated MISR versus TROPOMI plume heights are shown in (c), together with the statistics of the line of best fit (correlation coefficient $R$, slope $s$, as well as mean difference $\pm$ standard deviation in km). Five example profiles where simulated MISR and TROPOMI plume heights agree well are shown in d-h, and four example profile where the are significant differences are shown in i-l. The TROPOMI-OSSE and MISR-OSSE heights (d-l) are plotted with a $PM_{2.5}$ offset simply for visualization.



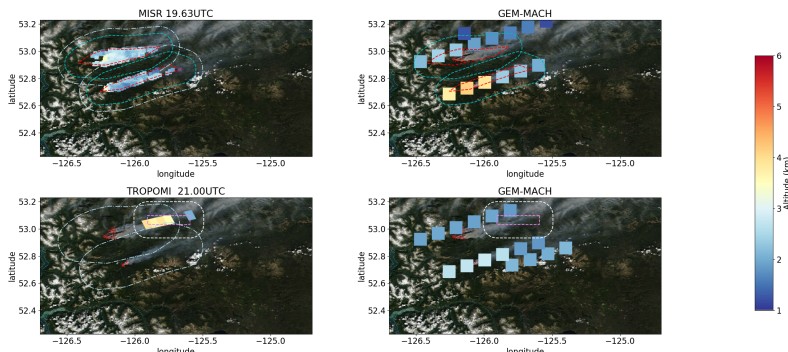

**Figure 2.** Example of two fire plumes on 6 August 2018 in British Columbia, Canada (approximately 56°N, 126°W). The color scheme illustrates the altitude of the plume (a) as observed by MISR at 19:38 UTC, (b) modeled in GEM-MACH using CFFEPS at 19:38 UTC, (c) observed by TROPOMI at 21:00UTC, and (d) modeled by GEM-MACH using CFFEPS at 21:00 UTC. The dashed lines outline the shape of the plume as used for the comparison (see text for further details). (Underlying VIIRS images obtained from NASA Worldview (https://worldview.earthdata.nasa.gov/)

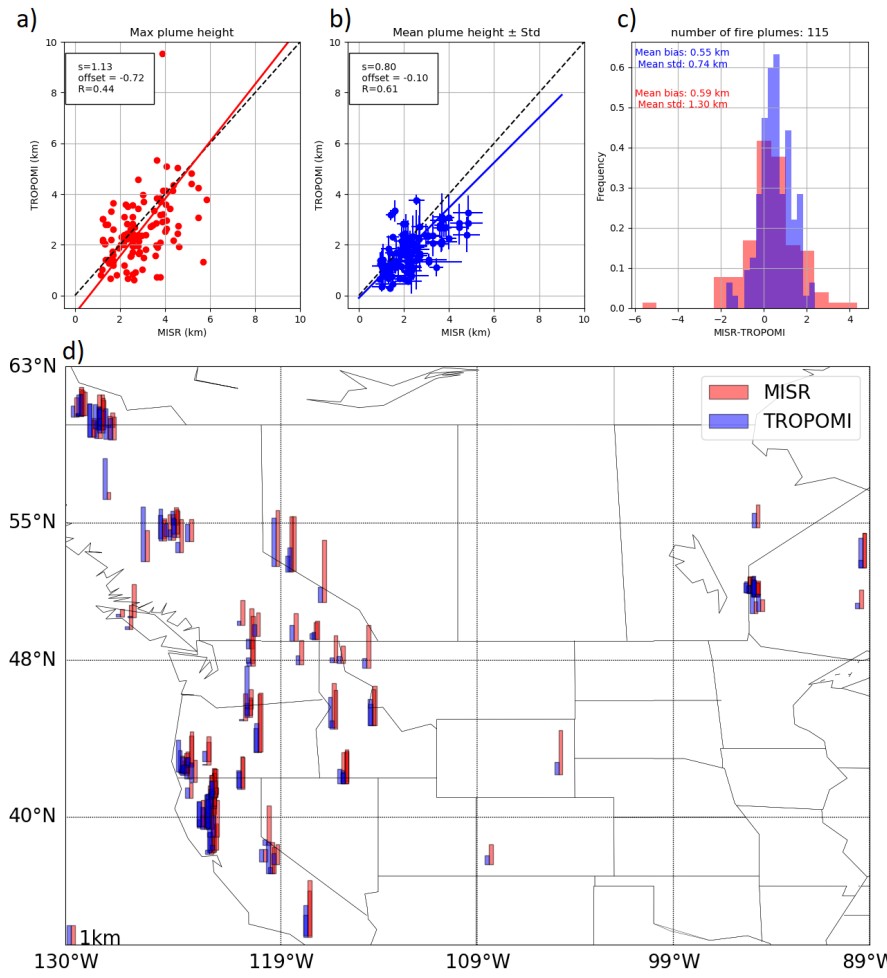

**Figure 3.** TROPOMI-MISR plume height comparison. In total 115 fire plumes were compared (all between June–August 2018) in terms of (a) maximum plume height and (b) mean plume height within one wildfire plume, together with the statistics of the line of best fit (correlation coefficient, $R$ and slope, $s$). (c) shows the histogram for the differences in plume height (MISR-TROPOMI) for the maximum (blue) and mean (red) plume height. (d) is a map showing the regional distribution of those fires with the mean plume height (above ground level) for TROPOMI (blue, left bar) and MISR (red, right bar).

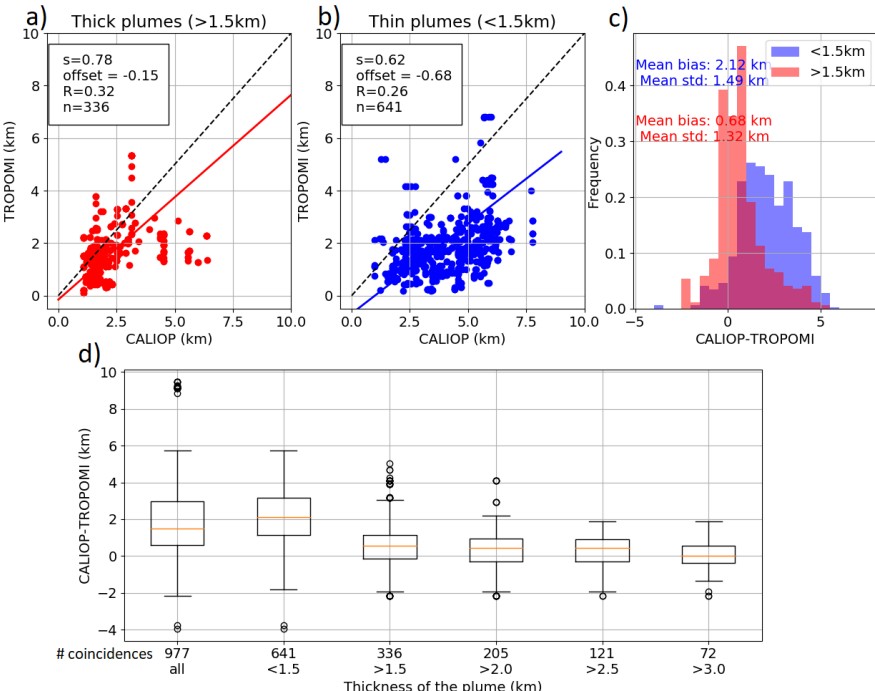

**Figure 4.** CALIOP-TROPOMI comparison for (a) thick plumes (>1.5 km) and (b) thin plumes (<1.5 km), together with the statistics of the line of best fit (correlation coefficient, $R$, slope, $s$, and number of observations, $n$). The plume thickness is determined by the CALIOP top and base plume height. (c) shows the histogram for the differences in plume height (CALIOP-TROPOMI) for thick (red) and thin (blue) plumes. (d) shows the statistics for different plume thickness filters.



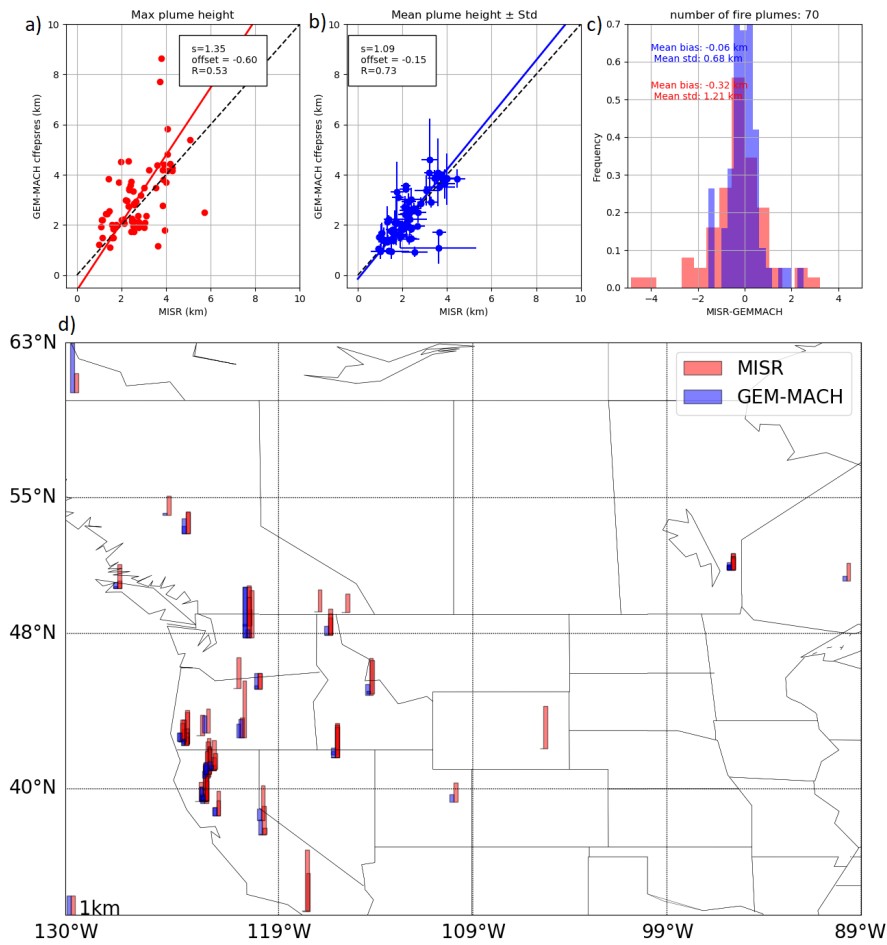

**Figure 5.** Modelled plume height versus MISR-derived plume height (a-c). In total 70 fire plumes were compared (all between June-August 2018) in terms of (a) maximum plume height and (b) mean plume height within one wildfire plume, together with the statistics of the line of best fit (correlation coefficient, $R$ and slope, $s$). (c) shows the histogram for the differences in plume height (MISR-model) for the maximum (blue) and mean (red) plume height. (d) shows the regional distribution of plume heights above ground level for MISR (red, left bar) and CFFEPS (blue, right bar).





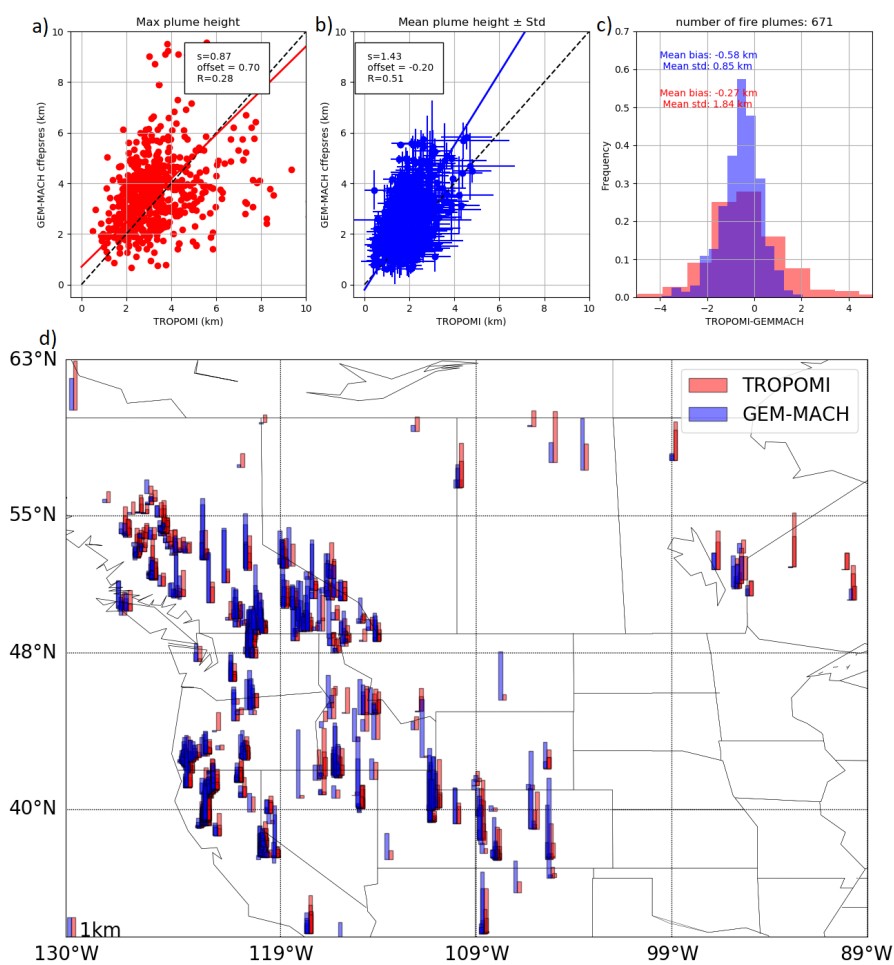

**Figure 6.** Same as Fig. 5, but for the TROPOMI-model comparison.