# Peer review of "The 2018 fire season in North America as seen by TROPOMI: aerosol layer height inter-comparisons and evaluation of model-derived plume heights"

_Atmospheric Measurement Techniques, 2019_

## Referee Comment (RC1) · Anonymous Referee #1 · 16 Nov 2019

1. Abstract, Conclusions, and Introduction paragraph 3. Recently a third global plume-height product has been created. It is a thermal technique, similar to one used for volcanic plumes in the past, and has been applied to MODIS. The reference is:

Lyapustin, A., Y. Wang, S. Korkin, R.A. Kahn, and D. Winker, 2019. MAIAC thermal technique for smoke injection height from MODIS. IEEE Geosci. Remt. Sens. Lett., doi: 10.1109/LGRS.2019.2936332.

2. Introduction, P2, lines 30-31. Also Section 6, P15, lines 4-6. MISR provides global coverage about once per week (about every 8 days near the equator, every 2 days near the poles). CALIPSO covers effectively 10-4 of the global surface, once every 16

days. This difference could be made clearer.

3. Introduction, P3, lines 6-10. MISR stereo heights have also been validated against ground lidars.

4. Section 2.1, P4, lines 22-23. I don't understand why a different (better) solar spectrum would be applied to the OFFL product than to the NRTI product. Once you have the better spectrum, can't it be used for the NRTI product too?

5. Section 2.1, P4, lines 25-26. As described, the "quality flag" sounds more like a plume detection flag; if so, this might be a better description. Have you evaluated its actual quality, e.g., by using the MODIS FRP product?

6. Section 2.2, P5, line 2. Martonchik et al. (2004) did not evaluate the MISR plume height products. The main references for this product would be Muller et al. (2002) and Moroney et al. (2002).

7. Section 2.2, P5, line 5. MISR actually has a standard stereo-height product, which is described in Muller et al. (2002) and Moroney et al. (2002). It runs on all the MISR data, and produces a reflectance-layer-reference-altitude, but does not call out aerosol plumes explicitly.

8. Section 2.2, P5, line 17. The narrow MISR swath limits the frequency of global coverage.

9. Section 2.3, P5, lines 27-28. The CALIOP "swath" is really a curtain, having a width of ∼100m, not several km. The data are usually averaged to several kilometers, but only along-track.

10. Section 2.3, P5, lines 30-31. Here you are using the CALIOP aerosol classification scheme, for which the key reference is: Omar, A.H., et al., 2009. The CALIPSO Automated Aerosol Classification and Lidar Ratio Selection Algorithm. J. Atm. Oce. Tech. 26, pp1994-2014, doi: 10.1175/2009JTECHA1231.1.

11. Section 2.4, P6, line 5. Small fires are also missed often by FRP, as well as those under heavy smoke plumes.

12. Section 3.1, P7, line 23. For MISR, the contrast is assessed at a spatial scale of 1.1 km, which probably provides a lot more of the plume vertical structure than the model simulation – in particular, more extreme height maxima and minima.

13. Section 3.1, P8, line 2. Note that these are very large indices of refraction, both real and especially imaginary. Might apply to BC near source, but probably not hydrated or aged smoke particles.

14. Section 4.2, P11, lines 29-30. CALIOP samples a curtain, so the data can be aggregated along-track to 5 km, but the cross-track width is still $\sim$ 100m. There is nothing you can do about this, but it is worth noting that the sampling footprints of CALIOP and TROPOMI are still quite dissimilar.

15. Section 4.2, P12, lines 14-15 and Fig. 4. Here you might emphasize that by "thick," you mean geometrically thick, and not optically thick. One would expect the differences in sampling among methods to be minimized for optically thick, geometrically thin plumes.

16. More generally, it might be helpful to identify explicitly the goal of the model and measurement comparisons in Sections 4 and 5. One would expect differences, due to different spatial and temporal sampling, as well as sensitivity to optical depth and optical depth vertical distribution, among the measurements. The model assumptions contribute to differences among the simulations and with the measurements. So this is not really a "validation," as these could all be "correct" in the context of what they measure or simulate. Rather, I think you are exploring the sensitivity of the "plume height" result to different plume properties, measurement techniques, and modeling assumptions. As such, I find most useful the conclusions presented where you interpret the differences in terms of attributes of the derivation methods and plume properties.
17. Section 6, P15, lines 18-19. I'm wondering whether the "exact plume height" is really well defined when there are multiple layers.

18. Section 6, P15, lines 19-20. Actually, most aerosol plumes are not uniform in optical thickness, and when multiple layers are present, they rarely cover exactly the same area. As such, MISR will often pick up multiple layers, not in a single 1.1 km pixel, but over the plume area imaged by the instrument.

---

## Referee Comment (RC2) · Anonymous Referee #2 · 3 Dec 2019

The paper presents the first analysis of TROPOMI ALH retrieval. OSSE is conducted to guide the analysis, which is excellent. This reviewer would recommend moderate revision of the paper to shed some light of surface albedo that may affect the ALH accuracy, and to provide somewhat closure between the past theoretical error analysis of ALH and the finding in the paper.

Datasets. What is the data resolution and range of TROPOMI ALH? In other words, in the retrieval, is ALH data continuous or in discrete values at different pressure level? Can ALH be 0.5 km or lower?

GEM-MATCH A few sentences describing how GEM-MACH estimate the injection

height can be insightful. Is the fire radiative power used in CFFEPS?

OSSE TROPOMI flume heights P8, L20. Using wavelength (instead of wavenumber) is suggested here. In addition, it is noted that TROPOMI uses the spectral fitting to derive ALH, not a simple ratio. In contrast, Xu et al. (2017, 2019, already cited in the manuscript) used the ratio.

Section 4.1. Some discussion about the reasons for MISR vs. modeled plume height difference can be helpful. Note, most satellite-based fire products provide only pixel-based FRP, where the plume rise model should use FRP over the fire area (not pixel area). This paper might be useful here to interpret the difference. Peterson et al., 2014, Quantifying the potential for high-altitude smoke injection in North American boreal forest using the standard MODIS fire products and sub-pixel-based methods, JGR.

Section 4.2 and results: There are multiple times, 'thin' layer is mentioned. How the thin layer is defined? By optical depth or geometric thickness? Past work has shown that O2-A type of ALH retrieval should be sensitive the high aerosol plumes provided a moderate value of AOD. The appendix in Xu et al. (2019) provides the ALH error estimates for different AOD and different ALH. It shows that retrieval is most sensitive to ALH at 2 km, and should be good to provide ALH from 1 – 8 km with retrieval error of less than .5km for AOD of 0.4 over dark surfaces. Anyhow, these past analyses should be helpful to interpret the physics behind the finding here. Afterall, past work have done several case studies to evaluate the ALH retrieved from O2 band (although not from TROPOMI). It is shown that the retrieval error can be affected not only AOD and ALH, but also by surface albedo. It might be interesting to stratify the ALH differences by surface albedo. As surface albedo increases, the ALH retrieval error can be large. Some comparison and contrasting of the results here with the results in the literature can be more revealing.

Summary L25-28, P15. Is your finding from the real data more or less consistent with the theoretical error analysis in Xu et al. (2019)?

L5-15, P16. Again, surface albedo is briefly mentioned and discussed here. It might be nice to sort the ALH evaluation by surface albedo. In addition, it is worth mentioning that for thick plumes at the surface, ALH retrieval is expected to have large errors. The analysis presented in the papers show the retrieved ALH is at least 1 km above the surface. There are also cases where TROPOMI ALH is consistent with CALIOP for high and think plumes (Fig. 4b). In other words, in both abstracts and conclusion, it is worth mentioning that the TROPOMI ALH has some success in retrieving high plumes up to 8 km (in addition to that the most accurate retrievals are for thickness plumes from 1-4.5 km).

---

## Referee Comment (RC3) · Anonymous Referee #3 · 17 Dec 2019

This manuscript presents the first analysis and evaluation of smoke injection heights retrieved from TROPOMI, using fires in North America from June to August 2018. The Authors compare the TROPOMI smoke height retrievals with MISR and CALIOP observations and plume heights derived from the CFFEPS. The manuscript presents results that are of interest to the readers of Atmos Meas Tech and the scientific community overall. As the Authors highlight, TROPOMI offers an additional smoke height plume product with higher spatial and temporal resolution than MISR and CALIOP, and with almost near real time availability. These characteristics are valuable for the modelling community, to forecast air quality impacts and for aviation safety, for example. I have added some comments and notes that will help improve the manuscript; I hope the

Authors consider them during the revision process.

*Introduction. The discussion about differences between the satellite needs to be clearer. I agree MISR and CALIOP are different instruments and use different methods to retrieve smoke heights. For example, MISR has a swath of 380 km common to all cameras, and global coverage is obtained every 9 days at the Equator and every 2 days at the poles. CALIOP swath is about 70 m wide, not kilometres as state in the manuscript, and this provides a global coverage every 16 days. However, they are at the same time complementary as they observe fires at different times and CALIOP is able to retrieve smoke from optically thinner plumes, whereas MISR offers a larger sample size, near the fire source.

*Page 3-Line 9. The planetary boundary layer tends to be higher later in the afternoon and that may contribute to the difference between MISR and CALIOP smoke heights.

*Page 4- Line 25. I don't understand the TROPOMI quality flag. Does this flag provide an indication of retrieval quality, or is it simple to define if there is a smoke plume retrieved?

*Page 5 Line 31. Why do you consider CALIOP aerosol plumes with polluted dust? The evaluation is for 'smoke' plumes.

*Page 6 Line 5. MODIS can also miss fires under high dense smoke.

*Page 6 Line 5. Do you use all MODIS thermo anomalies pixels or only those pixels with some confidence level that indicate active fire?

*Page 6- GEM-MACH. It is not clear to me what type of smoke injection height scheme CFFEPS uses. A brief description indicating the parameterization and key drivers will really help.

*Page 11 Line 27. Again, CALIOP profiles are selected with smoke and polluted aerosols (aerosols, not dust?).

*Page 12 Line 10. How does your definition of CALIOP smoke height (method 3) differ/compare from Huan et al., (2015) and Gonzalez-Alonso et al. (2019)?

*Page 12 Line 14. How do you define 'thick' and 'thin' plumes? Is it by size or by density?

*There is a Table S1 (Supplementary Materials), but it is not referenced within the manuscript

*The Authors mention the near-real time smoke plume height retrievals, but there is not mention within the text where the TROPOMI smoke height can be downloaded. A refence will be very useful for the readers.

References Gonzalez-Alonso, L., Val Martin, M., and Kahn, R. A.: Biomass-burning smoke heights over the Amazon observed from space, Atmospheric Chemistry and Physics, 19, 1685–1702, https://doi.org/10.5194/acp-19-1685-2019, https://www.atmos-chem-phys.net/19/1685/2019/, 2019.

Huang, J., Guo, J., Wang, F., Liu, Z., Jeong, M.-J., Yu, H., and Zhang, Z.: CALIPSO inferred most probable heights of global dust and smoke layers, J. Geophys. Res.-Atmos., 120, 5085–5100, 2015.

---

## Author Comment (AC1) · 31 Jan 2020

We would like to thank reviewer #1 for his/her reviews. We addressed each comment below and highlighted our answers in red, the referee's comments are black.

1. Abstract, Conclusions, and Introduction paragraph 3. Recently a third global plumeheight
product has been created. It is a thermal technique, similar to one used for volcanic plumes in the past, and has been applied to MODIS. The reference is: Lyapustin, A., Y. Wang, S. Korkin, R.A. Kahn, and D. Winker, 2019. MAIAC thermal technique for smoke injection height from MODIS. IEEE Geosci. Remt. Sens. Lett., doi: 10.1109/LGRS.2019.2936332.

We were unable to implement the MODIS plume heights, since our paper was submitted before the Lyapustin et al. paper was published. We have now included references to it in the introduction.

The MODIS plume heights are not a truly global product, since only plume heights are only valid near hotspots and exclusively for fire plumes (others should be filtered, see Lyapustin et al. Conclusions: "To exclude
the transported smoke and ensure good quality of retrievals, we currently recommend to use Ha within ±75–150 km from the detected thermal hotspots as reported in the MAIAC quality assurance (QA) flag in the MCD19A2 product."
Thus, we felt the sentence in our abstract "Before the launch of TROPOMI, only two other satellite instruments were able to observe aerosol plume heights globally, MISR and CALIOP." and in the conclusions "The only satellites that could globally observe plume heights before the launch of TROPOMI were MISR and CALIOP." are still valid.

We included a short discussion of the MODIS plume height in the introduction:

"Very recently another plume height product has been created from MODIS observations, utilizing a thermal contrast technique (Lyapustin et al., 2019). These estimates are available globally, however, they are limited to plume heights near thermal hotspots."

2. Introduction, P2, lines 30-31. Also Section 6, P15, lines 4-6. MISR provides global coverage about once per week (about every 8 days near the equator, every 2 days near the poles). CALIPSO covers effectively 10-4 of the global surface, once every 16 days. This difference could be made clearer.

We have changed the sentence on p.2 accordingly.
From:
"However, these two instruments have the disadvantage of very limited coverage where most fires are missed […]."
To:
""However, these two instruments have the disadvantage of very limited coverage where most fires are missed […]; MISR provides global coverage about once per week

(8 days near the equator and every two days near the poles) and CALIPSO provides global coverage about every 16 days."

And on p. 15:
"those two satellites have a narrow-swath with a global coverage every week and 16 days, respectively."

3. Introduction, P3, lines 6-10. MISR stereo heights have also been validated against ground lidars.

We included the following sentence:
"Caliop and (standard) MISR plume heights have also been validated with ground-based lidars (e.g. Moroney et al., 2002; Naud et al., 2004; Kim et al, 2008; Tao et al.,2008)."

References:

Moroney, C., R. Davies, and J.-P. Muller (2002), MISR stereoscopic image matchers: Techniques and results, IEEE Trans. Geosci. Remote Sens., 40, 1547– 1559.

Naud, C., J. Muller, M. Haeffelin, Y. Morille, and A. Delaval (2004), Assessment of MISR and MODIS cloud top heights through intercomparison with a back-scattering lidar at SIRTA, Geophys. Res. Lett., 31, L04114, doi:10.1029/2003GL018976.

Kim, S.-W., Berthier, S., Raut, J.-C., Chazette, P., Dulac, F., and Yoon, S.-C.: Validation of aerosol and cloud layer structures from the space-borne lidar CALIOP using a ground-based lidar in Seoul, Korea, Atmos. Chem. Phys., 8, 3705–3720, https://doi.org/10.5194/acp-8-3705-2008, 2008.

Tao, Z., McCormick, M. & Wu, D. A comparison method for spaceborne and ground-based lidar and its application to the CALIPSO lidar. Appl. Phys. B 91, 639 (2008) doi:10.1007/s00340-008-3043-1

4. Section 2.1, P4, lines 22-23. I don't understand why a different (better) solar spectrum
would be applied to the OFFL product than to the NRTI product. Once you have
the better spectrum, can't it be used for the NRTI product too?

At the time of the NRTI processing the different (better) irradiance spectrum is not available.
NRTI retrievals are delivered within three hours of sensing, so only data available at that time can be used. In the OFFL processing more data are available, such as an

irradiance measurement closer to, but after, the radiance measurement. This is used precisely as the reviewer suggest (to use it in the retrieval then too), but is then called OFFL data. The differences between the data streams are not really important for this paper. The OFFL data was used, which is the best choice, when the time delay is not an issue.

5. Section 2.1, P4, lines 25-26. As described, the "quality flag" sounds more like a plume detection flag; if so, this might be a better description. Have you evaluated its actual quality, e.g., by using the MODIS FRP product?

We have changed the description to the following to make the meaning of the quality flag a little clearer:
"In general, the OFFL product should perform better and is a better choice if timeliness is not an issue. Here, we evaluate the OFFL version only, as the NRTI version was not available for the time period that we investigated. As a first indication, the quality of each successful ALH retrieval is indicated by a quality assurance values (qa_value). If the input data or measurement configuration becomes close to a predefined limit, first the qa_value is lowered, if another limit is crossed, the pixel is filtered. E.g. all pixels with a solar zenith angle below 60º should have a good quality retrieval. However, for SZA > 60º the curvature of the Earth and the long photon path through the atmosphere may compromise a good retrieval. Above 75º, no retrievals are attempted. However, between 60º <SZA<75ºthe retrieval is performed, but the qa_values are lowered to 20%, to indicate to the user to use caution. This is done for all pixels with a (small) cloud fraction (qa lowered by 50%), small AAI (50%), high surface roughness (50%), and within sunglint and south Atlantic anomaly regions (50%).
Apart from the quantitative layer height, the quality flag provided alongside can be useful by itself, e.g. to locate and identify the presence of aerosol plumes and its vertical shape."

The quality flag cannot be evaluated with the MODIS FRP for several reasons:
A TROPOMI AER_LH may not be detected near a MODIS hotspot because:
- the plumes are typically downwind from the fires
- there might be clouds that interfere with the signal
- the SZA is too high to retrieve the TROPOMI AER_LH

A MODIS hotspot may not be near a TROPOMI AER_LH observation because:
- the TROPOMI AER_LH is not restricted to fire plumes, but can be from any source (volcanoes, dust, …)
- also some plume may be transported a long distance and can still be picked up by TROPOMI

6. Section 2.2, P5, line 2. Martonchik et al. (2004) did not evaluate the MISR plume height products. The main references for this product would be Muller et al. (2002) and Moroney et al. (2002).

We changed the references as suggested. Thank you for pointing this out and correcting this.

7. Section 2.2, P5, line 5. MISR actually has a standard stereo-height product, which is described in Muller et al. (2002) and Moroney et al. (2002). It runs on all the MISR data, and produces a reflectance-layer-reference-altitude, but does not call out aerosol plumes explicitly.

We have changed the following sentence in the manuscript, from:
"The plume height is not a standard product of MISR"
To:
"An operational MISR cloud-top product is available, however, the operational algorithm uses fixed-parameters that are applied to all scenes equally (Muller et al., 2002; Nelson et al., 2013). Instead, the plume height used here is not a standard product of MISR…"

8. Section 2.2, P5, line 17. The narrow MISR swath limits the frequency of global coverage.

We have changed the sentence as suggested, from:
"..(1) the swath limits the global coverage,.."
To:
"…(1) the narrow MISR swath limits the frequency of global coverage,…"

9. Section 2.3, P5, lines 27-28. The CALIOP "swath" is really a curtain, having a width of _100m, not several km. The data are usually averaged to several kilometers, but only along-track.

Thank you for pointing out this mistake.
We have changed the sentence from:
"…and has a very narrow swath width of just a few kilometres. In this study, we use the daytime aerosol layer product v4 (``Layer_Top_Altitude'', ``Layer_Base_Altitude'') (McGill et al., 2007; Vaughan et al., 2009) which provides the top and base height of aerosol layers detected (between the surface and 30 km) averaged to a 5 km horizontal resolution,…"
To:
"…and has a very narrow swath width of just a hundred meters. In this study, we use the daytime aerosol layer product v4 (``Layer_Top_Altitude'', ``Layer_Base_Altitude'') (McGill et al., 2007; Vaughan et al., 2009) which provides the top and base height of aerosol layers detected (between the surface and 30km) averaged over 5 km along the 100 m wide swath,…"

10. Section 2.3, P5, lines 30-31. Here you are using the CALIOP aerosol classification scheme, for which the key reference is: Omar, A.H., et al., 2009. The CALIPSO Automated
Aerosol Classification and Lidar Ratio Selection Algorithm. J. Atm. Oce. Tech.
26, pp1994-2014, doi: 10.1175/2009JTECHA1231.1.

Thank you we included the reference as suggested.

11. Section 2.4, P6, line 5. Small fires are also missed often by FRP, as well as those under heavy smoke plumes.

We modified the text to reflect this.
From:
"Note, that fires can potentially be missed due to cloud cover."
To:
"Note, that fires can potentially be missed for several reasons: due to cloud cover, under thick smoke plumes, as well as if the FRP signal is too low (e.g. small fires)."

12. Section 3.1, P7, line 23. For MISR, the contrast is assessed at a spatial scale of 1.1 km, which probably provides a lot more of the plume vertical structure than the model simulation – in particular, more extreme height maxima and minima.

Indeed, on a much finer resolution the minima and maxima would be more extreme. To reflect the resolution difference we averaged the MISR results to 10x10km (0.1x0.1deg) – the approximate resolution of the model - to be able to better compare the model and MISR plume heights. See text p. 11 l.16-18:
"To correct the impact of sensor resolution on the maximum plume height derived from a cluster of pixels in a given plume, the MISR pixels were averaged and binned on a 0.05x0.05 grid to approximately match the TROPOMI resolution."

13. Section 3.1, P8, line 2. Note that these are very large indices of refraction, both real and especially imaginary. Might apply to BC near source, but probably not hydrated or aged smoke particles.

The reviewer is correct that the real and imaginary parts of the refractive index will both decrease as the particle ages, but we are looking at fire plumes near the source. Kou (1996) (cited in the manuscript) found the value is 1.75 + 0.44i at 0% relative humidity [RH] for the complex refractive index of black carbon. This refractive index is unchanged up to 70% RH and is used by GEOS-CHEM. My value is for RH=99% is based on the assumption that there might be significant water from the combustion.

The thesis can be found here https://dalspace.library.dal.ca/handle/10222/55517; and further details can be found on p.12 of the thesis (p.32 if you use Adobe Reader's numbering).

Further, we realized that black carbon has a very high imaginary component in the refractive index. Other aerosols that might be part of a smoke plume is organic carbon (OC) which has a very low imaginary part – we used  1.36 + 0.001i (at RH=99%). Overall, we found there was not much difference between these two extreme cases of refractive index. The truth is probably a combination of BC and OC refractive index.

Many "MISR OSSE" plume heights were unchanged, and on average, we found the plume heights were 100m lower for OC than for BC.

We included the following in the manuscript to address the reviewer's comments, p.x l.x:

"…at 99 % relative humidity (which is expected near the fire source): 1.68+0.36i…"

And p.8 l.23-25:

"We have also estimated the plume height assuming organic carbon (OC) with a refractive index of 1.36 + 0.001i (at RH=99 %), and found negligible differences between the plume heights obtained assuming BC and OC refractive indices for most cases (see Fig. S1 and S2)."

And in the supplement:

"**MISR OSSE with different refractive indices**

The black carbon (BC) reflective index (1.68+0.36i, RH: 99%) has an extremely large imaginary part, different to the refractive index of organic carbon (OC) where the refractive index is 1.36 + 0.001i (at RH=99%).

Looking at these two extreme cases of refractive indices, little difference was found for the MISR OSSE plume heights, most plume heights were identical, see Fig.S1. Only for plume profiles with a small plume above a large plume we found differences: the estimates assuming BC returned the plume height of the upper plume whereas the estimates assuming OC picked up the lower plume (see Fig. S2).

[Figure]

Fig. S1: MISR OSSE plume height estimates assuming a refractive index of BC and OC. The plume heights are identical (or very similar) except for three cases (shown in Fig. S2) where the plume height assuming BC is higher than the plume height assuming OC.

[Figure]

Fig. S2: The three profiles that lead to different plume heights when assuming a refractive index for OC versus BC. This happened when there is a secondary, smaller top plume that is more reflected for BC aerosols.

"

14. Section 4.2, P11, lines 29-30. CALIOP samples a curtain, so the data can be aggregated along-track to 5 km, but the cross-track width is still ~ 100m. There is nothing you can do about this, but it is worth noting that the sampling footprints of CALIOP and TROPOMI are still quite dissimilar.

We added the following sentence in the text to address this:
"Note that the Caliop data is averaged to 5 km, however along a narrow swath (~100 m), differences could arise due to the different sampling."

15. Section 4.2, P12, lines 14-15 and Fig. 4. Here you might emphasize that by "thick," you mean geometrically thick, and not optically thick. One would expect the differences in sampling among methods to be minimized for optically thick, geometrically thin plumes.

We have changed the sentences as suggested, and included "geometrically".

We agree that probably geometrically thin but optically thick plumes should minimize the difference.

16. More generally, it might be helpful to identify explicitly the goal of the model and measurement comparisons in Sections 4 and 5. One would expect differences, due

to different spatial and temporal sampling, as well as sensitivity to optical depth and optical depth vertical distribution, among the measurements. The model assumptions contribute to differences among the simulations and with the measurements. So this is not really a "validation," as these could all be "correct" in the context of what they measure or simulate. Rather, I think you are exploring the sensitivity of the "plume height" result to different plume properties, measurement techniques, and modeling assumptions. As such, I find most useful the conclusions presented where you interpret the differences in terms of attributes of the derivation methods and plume properties.

We have changed the title to "The 2018 fire season in North America as seen by TROPOMI: aerosol layer height inter-comparisons and evaluation of model-derived plume heights" to remove the word "validation". The word "validation" is not mentioned in the manuscript.

The goal of the measurement comparisons in Sect. 4 and 5 can be found:
p. 1, l. 15-17
p. 3, l. 17-21

We further added a few sentences at the beginning of Sections 4 and 5 to highlight the purpose:
Satellite comparisons:

We added the following to Sect.4:
"As discussed in the previous section, there are fundamental differences between the plume heights observed by the different satellites. Here, the differences and correlation between the satellite plume height observations are discussed in terms of what is expected from the OSSE results and due to different observation times."

Model comparison:
The purpose of the model/satellite comparison is to evaluate if the model is "on the right track" or what it lacks. As discussed in the Sect. 5 and the conclusions, the modelled and observed plume heights correlate, however, especially over grassland, the model consistently overestimates the plume height, which is something that is helpful for the modelling community and something can be addressed in future releases of CFFEPS.

We added the following to Sect. 5:
"The modelled plume heights are compared to satellite observations with the aim to evaluate the modelled plume injection heights and to determine the strengths or weaknesses of the model."

17. Section 6, P15, lines 18-19. I'm wondering whether the "exact plume height" is really well defined when there are multiple layers.

We removed the word "exact" from the sentence.

18. Section 6, P15, lines 19-20. Actually, most aerosol plumes are not uniform in optical thickness, and when multiple layers are present, they rarely cover exactly the same area. As such, MISR will often pick up multiple layers, not in a single 1.1 km pixel, but over the plume area imaged by the instrument.

We have changed p.15, l.19-20, from:

"MISR on the other hand tends to respond to the upper aerosol layer, if there are any layers beneath MISR will not be able to pick this up."

To:

"MISR on the other hand tends to respond to the upper aerosol layer if multiple plumes overlap the same pixel, if there are any layers beneath MISR will not be able to pick this up. However, often multiple layers of plumes do not overlap exactly the exact same area, so MISR will likely sense the lower plume heights over the plume area imaged by the instrument."

---

## Author Comment (AC2) · 31 Jan 2020

We would like to thank reviewer #2 for his/her reviews. We addressed each comment below and highlighted our answers in red, the referee's comments are black.

The paper presents the first analysis of TROPOMI ALH retrieval. OSSE is conducted to guide the analysis, which is excellent. This reviewer would recommend moderate revision of the paper to shed some light of surface albedo that may affect the ALH accuracy, and to provide somewhat closure between the past theoretical error analysis of ALH and the finding in the paper.

Datasets.
What is the data resolution and range of TROPOMI ALH? In other words, in the retrieval, is ALH data continuous or in discrete values at different pressure level? Can ALH be 0.5 km or lower?

The vertical data resolution is continuous. It ranges from 1050-75 hPa. There is a data field that is fixed to the surface, but the original dataset, which can extend below the surface, is also available in the detailed results field. So, yes, the ALH can be lower than 0.5 km.

We have included the following sentence in the manuscript, Sect. 2.1:
"The vertical data resolution is continuous and ranges from 1050-75 hPa. "

GEM-MATCH A few sentences describing how GEM-MACH estimate the injection height can be insightful. Is the fire radiative power used in CFFEPS?

FRP is not used in CFFEPS, instead fire energy is estimated based on modelled fuel consumed and the estimated heat of combustion of dry fuel.  We included the following in the manuscript to describe how  the CFFEPS (GEM-MACH) injection height is estimated, in Sect. 2.5 GEM-MACH:

"Fire plume injection height in GEM-MACH is parameterized in the CFFEPS module with hourly modelled meteorology as detailed in Chen et al. (2019). The injection height is determined based on the balance of estimated plume buoyancy and the modelled environmental lapse rate at fire location. Total heat flux from fire is determined from modelled fuel consumed per area and the heat of combustion of dry wood fuel (Byram, 1959). The fraction of energy that enters the plume for convection is further parameterized based on thermodynamic energy balance accounting for heat lost to fuel, moisture, radiation, conduction and incomplete combustion. The hourly plume injection height is determined based on the dry adiabatic equilibrium of the buoyant plume and the modelled environmental lapse rate at fire location. "

OSSE TROPOMI flume heights P8, L20. Using wavelength (instead of wavenumber) is suggested here. In addition, it is noted that TROPOMI uses the spectral fitting to

derive ALH, not a simple ratio. In contrast, Xu et al. (2017, 2019, already cited in the manuscript) used the ratio.

As suggested, we changed the reported wavenumber to wavelengths throughout this paragraph.

Further, we have added the following sentence:
"Note that TROPOMI operational algorithm uses spectral fitting to retrieve AER_LH whereas a simple ratio has been used here, similar to Xu et al. (2017, 2019)."

Section 4.1. Some discussion about the reasons for MISR vs. modeled plume height difference can be helpful. Note, most satellite-based fire products provide only pixel based FRP, where the plume rise model should use FRP over the fire area (not pixel area). This paper might be useful here to interpret the difference. Peterson et al., 2014, Quantifying the potential for high-altitude smoke injection in North American boreal forest using the standard MODIS fire products and sub-pixel-based methods, JGR.

We included further description on how the model plume injection height is estimated, FRP is not used to determine the injection height with CFFEPS (inside the GEM-MACH model). See the comment above.

Section 4.2 and results: There are multiple times, 'thin' layer is mentioned. How the thin layer is defined? By optical depth or geometric thickness? Past work has shown that O2-A type of ALH retrieval should be sensitive the high aerosol plumes provided a moderate value of AOD. The appendix in Xu et al. (2019) provides the ALH error estimates for different AOD and different ALH. It shows that retrieval is most sensitive to ALH at 2 km, and should be good to provide ALH from 1 – 8 km with retrieval error of less than .5km for AOD of 0.4 over dark surfaces. Anyhow, these past analyses should be helpful to interpret the physics behind the finding here. Afterall, past work have done several case studies to evaluate the ALH retrieved from O2 band (although not from TROPOMI). It is shown that the retrieval error can be affected not only AOD and ALH, but also by surface albedo. It might be interesting to stratify the ALH differences by surface albedo. As surface albedo increases, the ALH retrieval error can be large. Some comparison and contrasting of the results here with the results in the literature can be more revealing.

Here, we mean geometrical thickness. We have added the word "geometrical" in front of thickness throughout the text. Also, p. 12 l.26 describes how the geometrical thickness is defined.

We have looked into the differences based on surface albedo and we have included Fig. 5e to the manuscript.

[Figure]

Showing that the differences between Caliop and TROPOMI increase with an increasing surface albedo.
We have added the following text to the manuscript, p.13, l. 19ff:
"Figure 4e shows that the differences between CALIOP and TROPOMI increase with increasing surface albedo, consistent with the idea that the TROPOMI retrieval algorithm is more sensitive over dark surfaces and possess smaller uncertainties (Sanders and de Haan, 2016; Xu et al., 2019)."

And in the conclusions:
From:
"… TROPOMI aerosol layer heights are more accurate for thicker plumes: the difference between the CALIOP and TROPOMI mid-plume height decreases and the correlation increases with increasing thickness of the plume and for a 3 km thick plume the average difference is only about 50 m."
To:
"…TROPOMI aerosol layer heights are more accurate for thicker plumes and over darker surfaces. As such, the difference between the CALIOP and TROPOMI mid-plume height decreases and the correlation increases with increasing thickness of the plume and for a 3 km thick plume the average difference is only about 50 m. Further, the differences between Caliop and TROPOMI increase with increasing surface albedo."

We can see something similar for MISR vs TROPOMI, however, not as many observations are available for the analysis, and thus we did not add it to the manuscript.

[Figure]

MISR mean plume height vs TROPOMI mean plume height. The differences are increasing for an increasing surface albedo. However, not as many observations are available and only 1-2 plumes have an albedo below 0.1 and above 0.2.

Summary L25-28, P15. Is your finding from the real data more or less consistent with the theoretical error analysis in Xu et al. (2019)?

We see that the differences between the instruments are larger for larger surface albedo, as mentioned in Xu et al., 2019, the error seems to increase with increasing surface albedo. Further, we looked into the AOD and found that we only have 7 cases (out of over 1000) that have AOD < 0.3. The differences for those are higher, however, it is only 7.

In terms of the surface albedo see the previous comment, as suggested by Xu et al., we can see that the TROPOMI plume height is closer to the plume height from MISR or Caliop for darker surfaces.

[Figure]

L5-15, P16. Again, surface albedo is briefly mentioned and discussed here. It might be nice to sort the ALH evaluation by surface albedo. In addition, it is worth mentioning that for thick plumes at the surface, ALH retrieval is expected to have large errors. The analysis presented in the papers show the retrieved ALH is at least 1 km above the surface. There are also cases where TROPOMI ALH is consistent with CALIOP for high and think plumes (Fig. 4b). In other words, in both abstracts and conclusion, it is worth mentioning that the TROPOMI ALH has some success in retrieving high plumes up to 8 km (in addition to that the most accurate retrievals are for thickness plumes from 1-4.5 km).

We have changed the following sentence in the abstract:

From:

"…our results show that the TROPOMI aerosol layer height is more accurate for thicker plumes and plumes below approximately 4.5 km."

To:

"… our results show that the TROPOMI aerosol layer height is more accurate for over dark surfaces, for thicker plumes and plumes between approximately 1-4.5 km."

We have changed the following sentence in the conclusions:

From:

"The TROPOMI plume heights seems more accurate for thicker and lower plumes plumes (<4.5 km altitude)."

To:

"The TROPOMI aerosl layer height seems to be successful in retrieving high plumes up to 8 km, the uncertainties seem reduced for thicker and lower plumes between 1-4.5 km altitude, as well as dark surfaces."

---

## Author Comment (AC3) · 31 Jan 2020

We would like to thank reviewer #3 for his/her reviews. We addressed each comment below and highlighted our answers in red, the referee's comments are black.

This manuscript presents the first analysis and evaluation of smoke injection heights retrieved from TROPOMI, using fires in North America from June to August 2018. The Authors compare the TROPOMI smoke height retrievals with MISR and CALIOP observations
and plume heights derived from the CFFEPS. The manuscript presents results that are of interest to the readers of Atmos Meas Tech and the scientific community overall. As the Authors highlight, TROPOMI offers an additional smoke height plume product with higher spatial and temporal resolution than MISR and CALIOP, and with almost near real time availability. These characteristics are valuable for the modelling community, to forecast air quality impacts and for aviation safety, for example. I have added some comments and notes that will help improve the manuscript; I hope theAuthors consider them during the revision process.

*Introduction. The discussion about differences between the satellite needs to be clearer. I agree MISR and CALIOP are different instruments and use different methods to retrieve smoke heights. For example, MISR has a swath of 380 km common to all cameras, and global coverage is obtained every 9 days at the Equator and every 2 days at the poles. CALIOP swath is about 70 m wide, not kilometres as state in the manuscript, and this provides a global coverage every 16 days. However, they are at the same time complementary as they observe fires at different times and CALIOP is able to retrieve smoke from optically thinner plumes, whereas MISR offers a larger sample size, near the fire source.

We have changed the sentence on p.2 accordingly.
From:
"However, these two instruments have the disadvantage of very limited coverage where most fires are missed […]."
To:
""However, these two instruments have the disadvantage of very limited coverage where most fires are missed […]; MISR provides global coverage about once per week (8 days near the equator and every two days near the poles) and CALIPSO provides global coverage about every 16 days."

We have changed the sentence on p. 5:
"…and has a very narrow swath width of just a few kilometres. In this study, we use the daytime aerosol layer product v4 (``Layer_Top_Altitude'', ``Layer_Base_Altitude'') (McGill et al., 2007; Vaughan et al., 2009)  which provides the top and base height of aerosol layers detected (between the surface and 30 km) averaged to a 5 km horizontal resolution,…"
To:

"…and has a very narrow swath width of just a hundred meters. In this study, we use the daytime aerosol layer product v4 (``Layer_Top_Altitude'', ``Layer_Base_Altitude'') (McGill et al., 2007; Vaughan et al., 2009) which provides the top and base height of aerosol layers detected (between the surface and 30km) averaged over 5 km along the 100 m wide swath,…"

And on p. 15:
"those two satellites have a narrow-swath with a global coverage every week and 16 days, respectively."

To address the last point, we added the following to the instruction to highlight that MISR and CALIOP are complementary because they observe fores at different times and with different methods:
"The time of observation and method used to determine the height of the plume is very different for these two instruments, making them complementary. Because the observation methods are different, it is important to…"

*Page 3-Line 9. The planetary boundary layer tends to be higher later in the afternoon and that may contribute to the difference between MISR and CALIOP smoke heights.

We have changed the sentence to:
"…difference of approximately 2 h can create additional challenges for comparing the plume heights, as the fire is expected to increase in intensity throughout the morning with the peak fire activity being in the early afternoon as well as changes in the planetary boundary layer that tends to be higher later in the afternoon…"

*Page 4- Line 25. I don't understand the TROPOMI quality flag. Does this flag provide an indication of retrieval quality, or is it simple to define if there is a smoke plume retrieved?
We have changed the description to the following to make the meaning of the quality flag a little clearer:
"In general, the OFFL product should perform better and is a better choice if timeliness is not an issue. Here, we evaluate the OFFL version only, as the NRTI version was not available for the time period that we investigated. As a first indication, the quality of each successful ALH retrieval is indicated by a quality assurance values (qa_value). If the input data or measurement configuration becomes close to a predefined limit, first the qa_value is lowered, if another limit is crossed, the pixel is filtered. E.g. all pixels with a solar zenith angle below 60º should have a good quality retrieval. However, for SZA > 60º the curvature of the Earth and the long photon path through the atmosphere may compromise a good retrieval. Above 75º, no retrievals are attempted. However, between 60º <SZA<75ºthe retrieval is performed, but the qa\_values is lowered by 80%, to indicate to the user to use caution. This is done for all pixels with a (small) cloud fraction (qa lowered by 50%), small AAI (50%), high surface roughness (50%), and within sunglint and south Atlantic anomaly regions (50%).

Apart from the quantitative layer height, the quality flag provided alongside can be useful by itself. Just this quality flag can be useful to locate and identify presence of aerosol plumes and its vertical shape."

*Page 5 Line 31. Why do you consider CALIOP aerosol plumes with polluted dust? The evaluation is for 'smoke' plumes.

We consider plumes containing either (or both) smoke or polluted dust. Looking at a few examples with clear contamination from fire plumes and we found that these can sometimes be classed as polluted dust rather than smoke; primarily we wanted to exclude all other aerosol types that cannot be from fires such as clean marine, dust, polluted continental, clean continental.
We changed the wording in the manuscript slightly, to:
"For the comparison between CALIOP and TROPOMI, only CALIOP plume heights over North America are retained and filter out (``Feature_Classification_Flags") the ones from clean marine, dust, polluted continental, clean continental, thus, only plume heights containing smoke or polluted dust were selected (we found that fire plume aerosols are classed as either smoke or polluted dust)."

Here is an example from 2018-08-20 over British Columbia, Canada that was filled with smoke from fires, in the aerosol classification this plume appears as smoke (3, orange) and polluted dust (5, brown).

[Figure]

*Page 6 Line 5. MODIS can also miss fires under high dense smoke.
We modified the text to reflect this.
From:
"Note, that fires can potentially be missed due to cloud cover."

To:
"Note that fires can potentially be missed for several reasons: due to cloud cover, under thick smoke plumes, as well as if the FRP signal is too low (e.g. small fires)."

*Page 6 Line 5. Do you use all MODIS thermo anomalies pixels or only those pixels with some confidence level that indicate active fire?

We use clustered MODIS thermal anomalies. We search for all thermal anomalies, with a confidence over 75%, the summed FRP of all thermal anomalies must be at least 1000 in the area (up to 5km radius) to be considered. We use these locations just as a starting point, of where potentially fires are.  This results in a bunch of potential fires each day. Then we look at these spots if MISR overpasses these areas and use MINX to trace the plume.
We use this list of dates and locations of fires for multiple things, however, we realized that for this study, we actually only used the MODIS fire anomalies to find potential fires in MINX, thus we have revised the MODIS section and moved it into the MISR description.
The text has been changed to:
"The MODIS thermal anomaly product (MOD14) (Giglio et al., 2003, 2006, 2016) is used here to locate the wildfires. We searched for clusters of thermal anomalies with a confidence of at least 75%, and a minimum summed FRP (within a 5km radius) of at least 1000. These locations were then used to search for plumes using the MINX package.  There are currently two MODIS instruments in space, on NASA's Terra and on NASA's Aqua satellites. Daytime measurements of Terra and Aqua are around 10:30 and 13:30 local time, respectively.  For the MINX analysis, we utilized the thermal anomalies from MODIS Terra. Note that fires can potentially be missed for several reasons: due to cloud cover, under thick smoke plumes, well as if the FRP signal is too low (e.g. small fires)."

*Page 6- GEM-MACH. It is not clear to me what type of smoke injection height scheme CFFEPS uses. A brief description indicating the parameterization and key drivers will really help.

We included the following in the manuscript to describe how  the CFFEPS (GEM-MACH) injection height is estimated, in Sect. 2.5 GEM-MACH:

"Fire plume injection height in GEM-MACH is parameterized in the CFFEPS module with hourly modelled meteorology as detailed in Chen et al. (2019). The injection height is determined based on the balance of estimated plume buoyancy and the modelled environmental lapse rate at fire location. Total heat flux from fire is determined from modelled fuel consumed per area and the heat of combustion of dry wood fuel (Byram, 1959). The fraction of energy that enters the plume for convection is further parameterized based on thermodynamic energy balance accounting for heat lost to fuel, moisture, radiation, conduction and incomplete combustion. The hourly plume injection

height is determined based on the dry adiabatic equilibrium of the buoyant plume and the modelled environmental lapse rate at fire location. "

*Page 11 Line 27. Again, CALIOP profiles are selected with smoke and polluted aerosols (aerosols, not dust?).

We meant dust, not aerosol and corrected it in the manuscript. See explanation above as to why we included polluted dust as well.

*Page 12 Line 10. How does your definition of CALIOP smoke height (method 3) differ/compare from Huan et al., (2015) and Gonzalez-Alonso et al. (2019)?

Similar to Gonzalez-Alonso et al. (2019), we use the CALIOP Level 2 version 4 data, with the difference that we only use daytime (closest to the TROPOMI overpass), and that we filter for smoke and polluted dust (instead of just smoke). Our approach is different to that from Huang et al. (2015), since we do not define the plume heights from aerosol extinction profile itself, but use the L2 averaged plume height product.

We included the following in the manuscript, p. 12 l.19-20:
"Similar to Gonzalez-Alonso et al. (2019), we use the top and plume base from the CALIOP L2 product (aerosol layer product v4), which are on a horizontal resolution of 5 km…"

*Page 12 Line 14. How do you define 'thick' and 'thin' plumes? Is it by size or by density?
Here, we are referring to geometrical thickness and included this ("geometrical") in the manuscript to make it clearer.
We also describe on p.12 l.26 how this geometrical thickness is defined. We have looked at optical thickness using the AOD within Caliop and TROPOMI, but couldn't find the same decreasing differences for increased AODs as found for the geometrical thickness. We believe this is likely due to the not very good AOD product from TROPOMI and Caliop.

*There is a Table S1 (Supplementary Materials), but it is not referenced within the Manuscript
We included a reference to the table as suggested, Sec. 2.5 p.7 l.1 :
"Differences between the operational and experimental version of GEM-MACH can be found in the supplement, Table S1."

*The Authors mention the near-real time smoke plume height retrievals, but there is not mention within the text where the TROPOMI smoke height can be downloaded. A refence will be very useful for the readers.

We included a data availability section to the manuscript, pointing to the locations where TROPOMI, Caliop, MISR, MODIS and the GEM-MAXH plume heights can be downloaded.

References Gonzalez-Alonso, L., Val Martin, M., and Kahn, R. A.: Biomass burning smoke heights over the Amazon observed from space, Atmospheric Chemistry and Physics, 19, 1685–1702, https://doi.org/10.5194/acp-19-1685-2019, https://www.atmos-chem-phys.net/19/1685/2019/, 2019.
Huang, J., Guo, J., Wang, F., Liu, Z., Jeong, M.-J., Yu, H., and Zhang, Z.: CALIPSO inferred most probable heights of global dust and smoke layers, J. Geophys. Res.-Atmos., 120, 5085–5100, 2015.

---

## Author Response (AR2)

**Dear Editor,**

**Thank you very much for finding these typos.**

**As suggested, we corrected these minor revisions in the manuscript, we highlighted our answers in bold.**

Comments to the Author:

The author(s) have provided a thorough response to the reviewers' comments and the revised manuscript includes all relevant changes.

However, I would like to ask you to make the following small corrections:

- Please use capital letter for CALIOP throughout the manuscript

**We have changed Caliop to CALIOP throughout the manuscript.**

- P.4, l13: 'The: vertical data resolution is continuous and ranges from 1050-75 hPa' -> I find this a bit confusing. I would suggest to say: 'The: vertical data resolution is continuous and values for ALH ranges from 1050-75 hPa'

As suggested, we changed the sentence to:

**"The vertical data resolution is continuous and values for the aerosol layer height (ALH) ranges from 1050-75 hPa."**

**Please note that we used aerosol layer height (ALH) instead of ALH, as ALH was not defined.**

- P17, l17: aerosl -> aerosol

**We correct this typo as suggested.**

**We further included the acronyms for AAI (absorbing aerosol index), and SZA (solar zenith angle) on page 5 l.1 and l. 5, respectively.**

**Thank you.**

**Debora Griffin et al.**

[revised manuscript text omitted]